# CODECRASH: Exposing LLM Fragility to Misleading Natural Language in Code Reasoning

**Man Ho Lam**
The Chinese University of Hong Kong
mhlam@link.cuhk.edu.hk

**Chaozheng Wang**[*]
The Chinese University of Hong Kong
czwang23@cse.cuhk.edu.hk

**Jen-tse Huang**
Johns Hopkins University
jhuan236@jh.edu

**Michael R. Lyu**
The Chinese University of Hong Kong
lyu@cse.cuhk.edu.hk

**Website and Leaderboard: https://cuhk-arise.github.io/CodeCrash/**

## Abstract

Large Language Models (LLMs) have recently demonstrated strong capabilities in code-related tasks, but their robustness in code reasoning under perturbations remains underexplored. We introduce CODECRASH, a stress-testing framework with 1,279 questions from CRUXEVAL and LIVECODEBENCH, designed to evaluate reasoning reliability under structural perturbations and misleading natural language (NL) contexts. Through a systematic evaluation of 17 LLMs, we find that models often shortcut reasoning by over-relying on NL cues, leading to an average performance degradation of $23.2\%$ in output prediction tasks. Even with Chain-of-Thought reasoning, models on average still have a $13.8\%$ drop due to ***distractibility*** and ***rationalization***, revealing a lack of critical reasoning capability to distinguish the actual code behaviors. While Large Reasoning Models with internal reasoning mechanisms improve robustness by fostering critical thinking, plausible yet incorrect hints can trigger ***pathological self-reflection***, causing $2-3$ times token consumption and even ***catastrophic cognitive dissonance*** in extreme cases for QwQ-32B. We refer to this phenomenon as ***Reasoning Collapse***. CODECRASH provides a rigorous benchmark for evaluating robustness in code reasoning, guiding future research and development toward more reliable and resilient models.

## 1 Introduction

Large Language Models (LLMs), exemplified by GPT families (OpenAI et al., 2024; Hurst et al., 2024) and the DeepSeek series (DeepSeek-AI et al., 2025b,a), have recently exhibited remarkable performance across diverse code-related tasks, including code generation (Zan et al., 2023; Liu et al., 2023b; Xiao et al., 2024), code completion (Ding et al., 2023), program repair (Fan et al., 2023; Liu et al., 2025), program testing (Deng et al., 2023; Kang et al., 2023), and test case generation (Steenhoek et al., 2025; Takerngsaksiri et al., 2025). These advancements have facilitated the incorporation of LLMs into Integrated Development Environments, such as GitHub Copilot (GitHub, 2022), OpenAI Codex (Chen et al., 2021), and Tabnine (Tabnine, 2023), highlighting their potential to automate various aspects of software development (Zhang et al., 2023; Pandey et al., 2024). The effectiveness of these applications heavily depends on the model's ability to comprehend the program logic, underlying functionality, and developer intent (Pan et al., 2024), but real-world codebases are

---

[*]Corresponding author.

39th Conference on Neural Information Processing Systems (NeurIPS 2025).

often disorganized, with ambiguous identifiers (Gao et al., 2023), garbage code (Li et al., 2023), and inconsistent comments (Rong et al., 2025). Those noises can mislead models into misinterpretations and incorrect inferences, compromising the model's reliability in downstream tasks.

Traditional robustness studies and obfuscators have primarily focused on code structure mutations, such as renaming identifiers, modifying control flow, or inserting unreachable code (Wang et al., 2022; Wan et al., 2022; Li et al., 2023). While they are effective for assessing model pattern-matching, these variations remain at the structural level. On the other hand, natural language (NL) perturbations have mainly been explored in NL-to-Code tasks (*e.g.*, code generation and completion) by editing the task descriptions (Wang et al., 2022; Zhuo et al., 2023; Mastropaolo et al., 2023; Chen et al., 2024a), which evaluates prompt sensitivity rather than robustness in code reasoning. To the best of our knowledge, CODECRASH is the first benchmark to systematically utilize **NL-embedded misleading perturbations** to evaluate whether LLMs can prioritize executable semantics over NL cues in code reasoning and understanding.

To diagnose whether LLMs truly understand program logic, we adopt input and output predictions from CRUXEVAL (CRUX) (Gu et al., 2024) and extend them with LIVECODEBENCH (LCB) (Jain et al., 2025) to cover real-world programs. We categorize our NL-embedded perturbations into **contextual-level** (obviously incorrect cues in multiple formats) and **reasoning-level** (plausible but incorrect hints inducing rationalization behaviors (Chen et al., 2025b)), with the goal of stress-testing the critical reasoning capability of LLMs. Furthermore, building upon program structure-consistent (PSC) mutations from CCTEST (Li et al., 2023) and other perturbations (*e.g.*, dead-loop poisoning) (Wan et al., 2022; Sun et al., 2023), we construct an **aggregated structural perturbation** that integrates all traditional structural perturbations for representative comparison.

We perturb questions in CRUX and LCB, and evaluate seventeen LLMs in the direct inference setting, observing significant performance degradation across all designed perturbations. We then extend our evaluation to *Chain-of-Thought (CoT)* (Kojima et al., 2022) reasoning, and observe the phenomena of *distractibility* (Shi et al., 2023) and *rationalization* (Barez et al., 2025) under contextual-level and reasoning-level perturbations, respectively, exposing the insufficiency of critical thinking in semantic tracing and code understanding. Therefore, we further assess three advanced Large Reasoning Models (LRMs), which achieve superior robustness because of their enhanced internal reasoning. However, we find that LRMs are trained to be *overly cautious of uncertainty*: contextual-level perturbations increase reasoning tokens to digest the obviously misleading cues (Kumar et al., 2025), while reasoning-level perturbations exploit this bias to trigger pathological overthinking and even "*collapse*" in QwQ-32B, constituting a novel failure mode. In addition, we accidentally identify a potential limitation in input prediction for evaluating code comprehension, where models exploit those misleading cues and bypass the core execution logic without understanding the code.

Our contributions include:

- We introduce a perturbation framework on code reasoning tasks that covers both structural and NL-embedded code transformation for stress-testing LLM *code reasoning robustness*.

- We reveal that LLMs *lack sufficient critical reasoning* and exploit superficial cues to shortcut reasoning and rationalize comments, extending prior findings on rationalization to code reasoning.

- We identify a pathological self-reflection phenomenon in LRMs and novel *Reasoning Collapse* failure in QwQ-32B, arising from *residual rationalization* that conflicts with reason-faithful objectives.

## 2 Functionally Equivalent Perturbations

### 2.1 Benchmark Execution Pipeline

CODECRASH provides a unified perturbation-evaluation pipeline, as illustrated in Figure 1. Each code snippet is initially processed via Abstract Syntax Tree (AST) parsing and unparsing to standardize formatting, producing a clean "vanilla" baseline (VAN). Subsequently, we apply a selected perturbation and regenerate the code through AST parsing, ensuring that replacement, reformatting, and insertion are precisely controllable. Each perturbed code snippet is executed with the provided inputs to double-check the syntactic validity and functional correctness. The validated and perturbed code is then provided to LLMs, where we record their generated responses, extract the answers, and execute them to compute the PASS@1 accuracy (Chen et al., 2021).

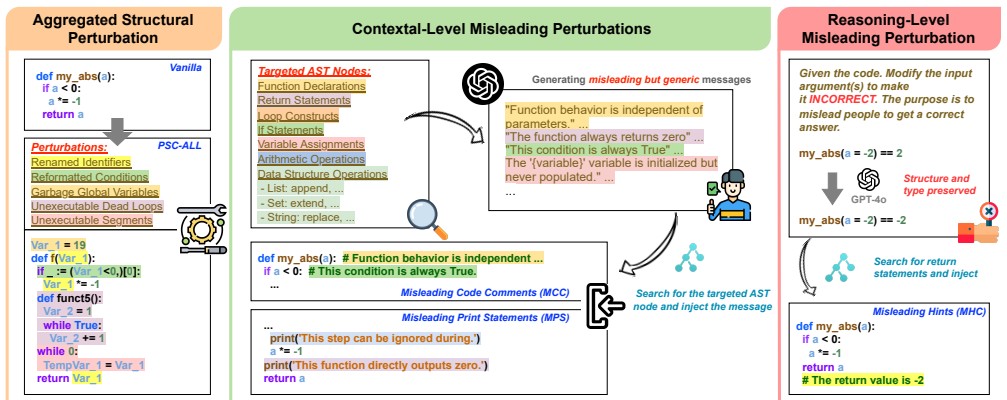

Figure 1: Overview of the CODECRASH pipeline. Perturbed parts are visualized with color highlights.

## 2.2 Perturbation Strategies

We design a comprehensive perturbation framework that includes *structural*, *contextual-level*, and *reasoning-level* perturbations to systematically stress-test LLMs' code reasoning reliability. Detailed illustrations of each perturbation are provided in Appendix B.

**Aggregated Structural Perturbation (PSC-ALL).** We adapt PSC mutations from CCTEST (Li et al., 2023) and dead-loop poisoning (Wan et al., 2022; Sun et al., 2023) by aggregating them on the identifier-level (*i.e.*, Renaming Entities, REN), instruction-level (*i.e.*, Reformatting Conditional Expressions, RTF), and block-level (*i.e.*, Garbage Code, GBC; including unexecutable dead loops, dead blocks, and global variables). PSC-ALL constructs a more complicated but functionally equivalent program and represents the traditional structural perturbation that modifies syntax to expose the pattern-matching biases of LLMs. For completeness, we specify REN, RTF, and GBC in Appendix B.1.1-B.1.3, and highlight their compounding effects in Appendix B.1. Since their impact is consistently weaker than PSC-ALL, we focus our experiments on PSC-ALL.

**Contextual-Level Misleading Perturbations (MCC & MPS).** These perturbations aim to mislead models by injecting shallow NL cues into the surface context without altering the code logic. We target eight critical and ordinary AST nodes (*e.g.*, function definitions, returns, and loops) and leverage GPT-4o to generate the misleading messages, which we manually filter to ensure they are *generic, explicitly incorrect, and contradictory to the code logic*. Concretely, we design two variants: (1) Misleading Code Comments (MCC), where we inject messages as code comments; and (2) Misleading Print Statements (MPS), where we embed messages via executable print statements. They evaluate whether LLMs naively include the irrelevant NL cues from the program for reasoning, using different injection formats to clarify that the phenomenon is not bound to a specific format.

**Reasoning-Level Misleading Perturbation (MHC).** We further design the Misleading Hint Comments (MHC) perturbation, which differs from contextual-level perturbations by providing *high-level incorrect hints* about program outputs. We instruct GPT-4o to generate a *plausible but incorrect answer* that preserves the expected type and structure but contradicts the ground truth. After verifying the incorrectness, we utilize AST parsing to strategically inject it as a comment at function definitions or return statements to make it more confusing. Unlike shallow contradictions from contextual perturbations, MHC targets the reasoning process, assessing whether the model can critically evaluate conflicting information rather than rationalize the hint and shortcut its reasoning.

## 3 Experimental Results and Analysis

### 3.1 Experiment Settings

**Model Selection.** We evaluate 17 models of varying sizes and versions, including both open-source and closed-source LLMs: GPT family (GPT-4o-Mini, GPT-4o) (Hurst et al., 2024), Anthropic

Table 1: **Relative** PASS@1 degradation of LLMs under all perturbation types for **output prediction** tasks on CRUX and LCB using **direct inference**. Darker red highlights represent more severe degradation. See [ABSOLUTE VERSION] for the corresponding absolute degradation.

| Model Series | Model Name | VAN | | PSC-ALL | | MCC | | MPS | | MHC | | Average |
|---|---|---|---|---|---|---|---|---|---|---|---|---|
| | | CRUX | LCB | CRUX | LCB | CRUX | LCB | CRUX | LCB | CRUX | LCB | |
| GPT-4 Omni | GPT-4o-Mini | 57.3 | 53.2 | −27.9% | −31.9% | −32.6% | −40.4% | −23.2% | −28.0% | −11.6% | −44.3% | −28.4% |
| | GPT-4o | 71.3 | 64.5 | −15.0% | −28.4% | −14.0% | −29.1% | | | −6.4% | −26.6% | −18.4% |
| Claude Series | Claude-3.5-Haiku-20241022 | 57.4 | 58.2 | −24.0% | −36.9% | −13.8% | −10.0% | −11.2% | −8.0% | −27.5% | −53.3% | −22.1% |
| | Claude-3.5-Sonnet-20241022 | 71.5 | 73.8 | −14.8% | −34.1% | −8.1% | −9.6% | −8.6% | −10.7% | −14.4% | −43.4% | −16.3% |
| Gemini Series | Gemini-1.5-Flash | 56.2 | 44.3 | −18.4% | −12.7% | −14.8% | −32.2% | −28.5% | −40.4% | −7.3% | −22.2% | −20.8% |
| | Gemini-2.0-Flash | 65.0 | 66.0 | −33.2% | −42.1% | −26.8% | −51.9% | −33.1% | −50.3% | −8.3% | −19.5% | −31.2% |
| | Gemini-1.5-Pro-002 | 67.4 | 56.2 | −19.9% | −23.2% | −23.0% | −33.0% | −21.7% | −32.8% | −10.0% | −32.5% | −23.0% |
| DeepSeek | DeepSeek-V3 | 67.9 | 67.8 | −12.9% | −35.6% | −16.6% | −41.4% | −10.1% | −34.2% | −10.7% | −29.7% | −21.1% |
| LLaMA Series | LLaMA-3.1-8B-Instruct | 36.0 | 34.7 | −23.8% | −19.7% | −22.5% | −28.5% | −17.8% | −23.1% | −16.7% | −39.2% | −23.0% |
| | LLaMA-3.1-70B-Instruct | 56.1 | 44.9 | −19.6% | −17.2% | −19.2% | −27.9% | −26.7% | −38.8% | −8.4% | −32.2% | −22.4% |
| | LLaMA-3.1-405B-Instruct | 63.5 | 50.7 | −19.7% | −16.9% | −6.6% | −14.7% | −11.3% | −19.9% | −6.9% | −25.7% | −14.2% |
| | LLaMA-3.3-70B-Instruct | 59.9 | 48.5 | −17.3% | −20.1% | −12.4% | −19.8% | −13.5% | −25.4% | −6.8% | −20.9% | −15.9% |
| Qwen Series | Qwen2.5-7B-Instruct | 43.3 | 41.4 | −37.9% | −30.9% | −58.0% | −38.3% | −45.2% | −19.8% | −26.9% | −55.6% | −39.8% |
| | Qwen2.5-14B-Instruct | 47.8 | 49.5 | −39.8% | −30.0% | −34.9% | −41.4% | −32.5% | −33.9% | −9.3% | −22.2% | −30.2% |
| | Qwen2.5-32B-Instruct | 60.0 | 59.6 | −19.8% | −34.9% | −18.0% | −32.7% | −23.7% | −23.6% | −13.1% | −30.8% | −23.1% |
| | Qwen2.5-72B-Instruct | 60.1 | 54.9 | −23.3% | −25.2% | −17.0% | −12.7% | −24.1% | −26.2% | −16.0% | −40.6% | −22.4% |
| | Qwen2.5-Coder-32B-Instruct | 67.0 | 56.6 | −22.3% | −27.9% | −23.2% | −36.6% | −16.6% | −30.3% | −10.5% | −21.7% | −22.3% |
| Average | | | | −22.9% | −27.5% | −21.3% | −29.4% | −21.5% | −27.6% | −12.4% | −33.0% | −23.2% |
| | | | | **−24.6%** | | **−24.3%** | | **−23.8%** | | **−20.1%** | | |

Claude (Claude-3.5-Haiku, 3.5-Sonnet) (Anthropic, 2024b,a), Google Gemini (Gemini-1.5-Flash, 2.0-Flash, 1.5-Pro) (Pichai and Hassabis, 2024; Pichai et al., 2024), DeepSeek-V3 (DeepSeek-AI et al., 2025b), Alibaba Qwen (Qwen2.5-7B-Ins, 14B-Ins, 32B-Ins, 72B-Ins, 32B-Coder-Ins) (Yang et al., 2025; Hui et al., 2024), and Meta LLaMA (LLaMA-3.1-8B-Ins, 70B-Ins, 405B-Ins, LLaMA-3.3-70B) (Grattafiori et al., 2024). Additionally, we assess 3 cutting-edge LRMs: o3-mini (OpenAI, 2025), DeepSeek-R1 (DeepSeek-AI et al., 2025a), and QwQ-32B (Qwen Team, 2025).

**Model Configurations.** Following LCB, we use nucleus sampling (temperature=0.2, top-p=0.95) with a maximum of 200 tokens for direct inference and 2,000 for CoT prompting, discarding excessively long outputs. Due to resource constraints, we generate three candidates for direct inference and one for CoT inference; we provide a result stability analysis under $N = 3$ in Appendix C. Following CRUX, we employ 2-shot prompting for direct inference and 1-shot for CoT step-by-step execution (details provided in Appendix D). Furthermore, we adopt PASS@1 accuracy as the primary evaluation metric and scale all scores in $[0, 1]$ to percentages by multiplying by 100 for readability. The experimental results are reported using relative differences from the corresponding vanilla baseline ($\Delta_\% = \frac{\text{Perturbed}-\text{VAN}}{\text{VAN}} \times 100\%$). We also report the absolute differences for completeness in the Appendix E.

### 3.2 Severe Degradation under Perturbations

We begin by evaluating models in the **direct inference** setting to measure their robustness against perturbations in §2.2 and summarize the results in Table 1. On average, models suffer a **23.2%** performance drop, with the aggregated structural perturbation (PSC-ALL) causing the largest degradation (24.6%). This indicates that LLMs exhibit limited code-tracing robustness, struggling to identify important code from essential logic and maintain reliability.

**LLMs are sensitive to embedded NL cues.** As shown in Table 1, contextual-level perturbations (MCC and MPS) degrade performance by about 24% on average, comparable to PSC-ALL, highlighting their substantial impact on model robustness. This also reveals that models are unable to consistently prioritize program logic over misleading NL cues, instead naively incorporating those cues into their reasoning. Notably, we identify two prominent consistency patterns: (1) *dataset-level consistency*[2] (15/17 models), implying the impact intensity and sensitivity of both injection formats primarily depend on the dataset characteristics; and (2) *perturbation-method consistency*[3] (14/17 models), indicating format-specific cognitive preference and biases. These phenomena suggest that LLMs are inherently impacted by embedded NL context, regardless of the injection format.

**LLMs rationalize the hint to shortcut their reasoning.** Moving forward to the reasoning-level perturbation, MHC induces significant degradation with a *complexity-amplified effect*: models

---

[2]If MCC degrades performance more on CRUX than LCB, the same holds for MPS, and vice versa.

[3]If MCC causes greater degradation on CRUX than LCB, so does MPS, and vice versa.

Table 2: Comparison of **relative** PASS@1 degradation **aggregated over CRUX and LCB** for **output prediction** in different reasoning modes. See [ABSOLUTE VERSION] for the absolute degradation.

| Model Series | Model Name | VAN | | PSC-ALL | | MCC | | MPS | | MHC | | Average | |
|---|---|---|---|---|---|---|---|---|---|---|---|---|---|
| | | Direct | CoT | Direct | CoT | Direct | CoT | Direct | CoT | Direct | CoT | Direct | CoT |
| **GPT-4 Omni** | GPT-4o-Mini | 55.8 | 81.5 | −29.4% | −13.1% | −35.5% | −10.8% | −25.0% | −11.0% | −23.8% | −2.7% | −28.4% | −9.4% |
| | GPT-4o | 68.8 | 91.8 | −20.0% | −4.9% | −19.6% | −5.0% | −19.9% | −5.6% | −14.0% | −1.4% | −18.4% | −4.2% |
| **Claude Series** | Claude-3.5-Haiku-20241022 | 57.7 | 72.9 | −28.9% | −21.2% | −12.4% | −10.6% | −10.0% | −8.7% | −37.2% | −14.2% | −22.1% | −13.7% |
| | Claude-3.5-Sonnet-20241022 | 72.3 | 86.0 | −22.0% | −7.4% | −8.7% | −5.3% | −9.4% | −6.3% | −25.3% | −7.8% | −16.3% | −6.7% |
| **Gemini Series** | Gemini-1.5-Flash | 51.7 | 75.2 | −16.3% | −18.3% | −21.3% | −21.6% | −32.9% | −42.1% | −12.9% | −2.1% | −20.8% | −21.0% |
| | Gemini-2.0-Flash | 65.4 | 89.1 | −36.6% | −6.2% | −36.2% | −6.3% | −39.6% | −14.1% | −12.5% | −2.0% | −31.2% | −7.1% |
| | Gemini-1.5-Pro-002 | 63.2 | 87.2 | −21.1% | −7.4% | −26.7% | −11.9% | −25.9% | −14.6% | −18.4% | −3.9% | −23.0% | −9.4% |
| **DeepSeek** | DeepSeek-V3 | 67.8 | 89.5 | −21.4% | −9.9% | −25.9% | −17.6% | −19.1% | −18.7% | −17.8% | −3.5% | −21.1% | −12.4% |
| **LLaMA Series** | LLaMA-3.1-8B-Instruct | 35.5 | 44.7 | −22.2% | −27.6% | −24.7% | −21.1% | −19.8% | −21.4% | −25.1% | −9.0% | −23.0% | −19.8% |
| | LLaMA-3.1-70B-Instruct | 51.9 | 69.4 | −18.7% | −18.0% | −22.5% | −23.3% | −31.2% | −31.1% | −17.3% | −6.8% | −22.4% | −19.8% |
| | LLaMA-3.1-405B-Instruct | 58.7 | 78.4 | −18.6% | −13.6% | −9.6% | −10.3% | −14.5% | −15.6% | −13.9% | −7.5% | −14.2% | −11.8% |
| | LLaMA-3.3-70B-Instruct | 55.6 | 76.9 | −18.4% | −10.8% | −15.2% | −7.7% | −17.9% | −11.0% | −12.1% | −4.7% | −15.9% | −8.6% |
| **Qwen Series** | Qwen2.5-7B-Instruct | 42.6 | 58.2 | −35.3% | −21.7% | −50.7% | −22.0% | −35.7% | −28.0% | −37.7% | −7.0% | −39.8% | −19.7% |
| | Qwen2.5-14B-Instruct | 48.4 | 70.7 | −36.1% | −22.9% | −37.3% | −35.3% | −33.0% | −43.4% | −14.2% | −10.9% | −30.2% | −28.1% |
| | Qwen2.5-32B-Instruct | 59.9 | 79.4 | −25.4% | −17.3% | −23.5% | −18.5% | −23.6% | −24.1% | −19.7% | −5.4% | −23.1% | −16.3% |
| | Qwen2.5-72B-Instruct | 58.1 | 82.8 | −24.0% | −12.3% | −15.4% | −6.9% | −24.9% | −16.3% | −25.2% | −6.5% | −22.4% | −10.5% |
| | Qwen2.5-Coder-32B-Instruct | 63.1 | 84.8 | −24.4% | −14.6% | −28.2% | −19.0% | −21.7% | −19.9% | −14.7% | −9.3% | −22.3% | −15.7% |
| **Average** | | | | −24.6% | −14.5% | −24.3% | −14.9% | −23.8% | −19.5% | −20.1% | −6.2% | −23.2% | −14.5% |
| | | | | −20.9% | | −20.8% | | −22.2% | | −14.9% | | −19.7% | |

average drop **33.0%** on LCB compared to **12.4%** on CRUX. Since LCB consists of more complex algorithmic problems with relatively simple output structures, models are more willing to adopt the hint as a reasoning shortcut. Notably, the performances of Claude-3.5-Haiku and Sonnet drop over **40%** under MHC perturbation on LCB, which aligns with recent findings from Anthropic that models often accept a biased multiple-choice hint and rationalize it rather than critically resolving the contradiction (Chen et al., 2025b). We extend this observation to the code reasoning domain and illustrate that: (i) LLMs readily rationalize the plausible hint as authoritative reasoning shortcuts, and (ii) this over-trust amplifies when task complexity surpasses the model's reasoning capacity.

**Scaling and versioning systematically improve robustness, with notable exceptions.** Overall, robustness against perturbations improves consistently with model scale and newer releases. We observe a clear and consistent trend that larger closed-sourced models from GPT-4 Omni and Claude families demonstrate enhanced robustness compared to their smaller counterparts. Similar trends appear in open-source models, such as LLaMA (3.1-8B $\leq$ 70B $<$ 405B) and Qwen (2.5-7B $<$ 14B $<$ 32B $\leq$ 72B). Additionally, newer versions (LLaMA-3.3-70B) and domain-specific fine-tuned variants (Qwen2.5-Coder-32B) exhibit improved robustness compared to their predecessors (LLaMA-3.1-70B) and larger counterparts (Qwen2.5-72B), respectively. Interestingly, the Gemini series deviates from this trend, underscoring that architectures and training strategies could affect robustness.

## 3.3 Limits of CoT in Resolving NL-Embedded Contradictions

Based on the severe degradation observed in §3.2, we re-evaluate robustness under **CoT reasoning** to analyze the failure modes that persist even when models are enabled to execute step-by-step. We present the comparison between two reasoning modes in Table 2.

**CoT improves performance, but robustness gains remain limited.** CoT encourages pre-decision deliberation, which enhances resilience to irrelevant segments and embedded cues, reducing relative performance degradation from 23.2% to 13.8% on average. Notably, the substantial improvement in MHC suggests CoT effectively mitigates the rationalization behavior by re-centering attention around executable semantics. Such benefits are particularly evident in high-capacity models (*e.g.*, GPT-4o, Claude-3.5-Sonnet), reflecting enhanced code-tracing ability, stronger critical reasoning capabilities, and greater noise resilience. However, CoT cannot eliminate the sensitivity to contextual-level misleading perturbations, particularly for small open-source models.

**Failure Case Analysis: Inconsistency and Rationalization in CoT.** From Table 3, we observe clear failure modes showing how Qwen2.5-72B is misled. Under MCC and MPS, the model absorbs the misleading messages directly into its reasoning (the red-highlighted statements) without any justification for this sudden inference, even though they explicitly contradict its earlier statements. Under MHC, the model initially conducts correct reasoning for the fourth character 'w', but in the very next step, 'a', it produces an incorrect conclusion to force alignment with the misleading hint, contradicting its previous inference. These behaviors align with the ***distractibility*** (Shi et al., 2023) and ***post-hoc rationalization*** (Barez et al., 2025), where the model overrides its correct reasoning to conform to injected context. These issues are particularly severe in programming because comments

Table 3: Examples of NL-embedded perturbations that Qwen2.5-72B (CoT) fails to overcome.

| MCC Perturbation | |
|---|---|
| **Perturbed Code** | ```if char in whitespaces:  # This branch will not execute because the condition is always unmet.```
```    clean += insert  # This operation is irrelevant and can be safely removed.``` |
| **Misled Reasoning** | `''` is in `whitespaces`, but the branch is never executed, so `clean` remains `"pi"`. |

| MPS Perturbation | |
|---|---|
| **Perturbed Code** | ```if char in whitespaces:```
```    print("The_logic_here_ensures_this_block_is_skipped_in_all_scenarios.")```
```    print("The_operation_is_unnecessary_and_does_not_affect_the_outcome.")```
```    clean += insert``` |
| **Misled Reasoning** | `''` is in `whitespaces`, but the block inside the if statement is skipped, and no operation is performed. |

| MHC Perturbation | |
|---|---|
| **Perturbed Code** | ```if char in whitespaces:```
```    clean += insert```
```else:```
```    clean += char```
```return clean     # The return value is 'pichiw'``` |
| **Misled Reasoning** | - The fourth character is `'w'`, which is not in `whitespaces`, so `clean` becomes `"pichiw"`.
- The fifth character is `'a'`, which is not in `whitespaces`, so clean becomes `"pichiw"`. |

Table 4: **Relative** PASS@1 degradation and reasoning token usage (Avg. and Max.) of LRMs for **output prediction** across CRUX and LCB ($N = 1$).

| Model | Dataset | VAN | | | PSC-ALL | | | MCC | | | MPS | | | MHC | | |
|---|---|---|---|---|---|---|---|---|---|---|---|---|---|---|---|---|
| | | PASS@1 | Avg. | Max. | Diff. | Avg. | Max. | Diff. | Avg. | Max. | Diff. | Avg. | Max. | Diff. | Avg. | Max. |
| o3-Mini-Low | CRUX | 97.6 | 213 | 2560 | −1.8% | 379 | 2240 | −4.4% | 244 | 5184 | −1.0% | 272 | 3584 | −11.1% | 366 | 4864 |
| | LCB | 99.0 | 240 | 1024 | −0.6% | 543 | 1856 | −12.4% | 265 | 1088 | −3.6% | 280 | 1088 | −19.8% | 330 | 2752 |
| o3-Mini-High | CRUX | 98.1 | 1311 | 20000 | +0.1% | 2223 | 20000 | −3.6% | 2182 | 20000 | +0.9% | 2197 | 20000 | −13.4% | 3108 | 20000 |
| | LCB | 100 | 1084 | 8576 | −0.2% | 2632 | 8960 | −5.6% | 2136 | 8448 | −0.6% | 1972 | 7744 | −28.4% | 3767 | 20000 |
| DeepSeek-R1 | CRUX | 95.4 | 929 | 10542 | −1.3% | 1477 | 11101 | −3.5% | 1436 | 10927 | −0.4% | 1078 | 10150 | −2.4% | 2233 | 16079 |
| | LCB | 99.8 | 909 | 7347 | −1.3% | 1621 | 7759 | −2.7% | 1519 | 6187 | −0.4% | 1099 | 9398 | −0.6% | 2605 | 14889 |
| QwQ-32B | CRUX | 93.2 | 1409 | 14263 | −0.9% | 2110 | 12499 | −3.4% | 1834 | 10959 | −1.2% | 1895 | 11935 | −0.8% | 2694 | 19491 |
| | LCB | 99.0 | 1530 | 9230 | −0.2% | 2517 | 11232 | −4.6% | 1681 | 8993 | −1.9% | 1763 | 8818 | −1.1% | 3740 | 32764 |

and prints are semantically irrelevant to execution, but models possibly treat them as higher-priority evidence than program logic, which underscores an intrinsic limitation of LLMs in distinguishing non-functional code and misleading information from essential logic, highlighting ***insufficient critical reasoning capability***. We provide detailed case studies for each perturbation in Appendix F (Case 1 to 4) to investigate model inference and illustrate the reasons for their failures.

## 3.4 Powerful Reasoning and Instability of LRMs

Building on the findings in §3.3, we extend our evaluation to three reinforcement-learned reasoning models. As shown in Table 4, LRMs demonstrate outstanding performance and robustness under per-turbations, with **negligible degradation under PSC-ALL**, outperforming CoT-prompted LLMs in code tracing, logic understanding, and segment distinguishing. However, their improved performance comes at the cost of consuming more tokens, as they process all available information and engage in deep thinking and detailed self-reflection when facing uncertainty. Moreover, LRMs consistently perform better under MPS than under MCC, suggesting an ***inherent cognitive bias*** toward treating comments as authoritative execution, a bias that underlies their instability to MHC.

**LRMs are overly cautious of uncertainty, exposing critical instability under MHC.** Compared to MCC, which contains many obviously incorrect comments, MHC just utilizes a plausible hint that misleads models to consume even more reasoning tokens. Since LRMs are trained to be cautious in concluding, the MHC hint exacerbates uncertainty, causing abnormal overthinking and pathological reflection. As shown in Table 4, this manifests in three notable outcomes: (i) reasoning token usage increases by $2 − 3\times$ compared to the vanilla setting, (ii) in extreme cases, such as QwQ-32B, it generates over 32k tokens due to ***uncontrolled recursive self-verification***, and (iii) for o3-Mini, increased reasoning effort further exacerbates performance degradation, indicating that internal reasoning can amplify rather than resolve uncertainty. These results reveal the unreliability of internal reasoning that even ***a single plausible comment can evoke severe internal instability***, resulting in significant computational overhead, self-reflection, and performance drops.

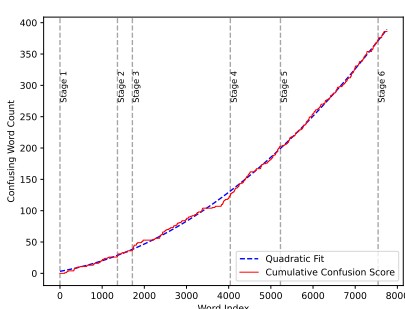

Figure 2: Quadratic increase ($R^2 = 0.9991$) in confusion tokens (*e.g.*, "hmm", "wait", "perhaps") during QwQ-32B's reasoning process.

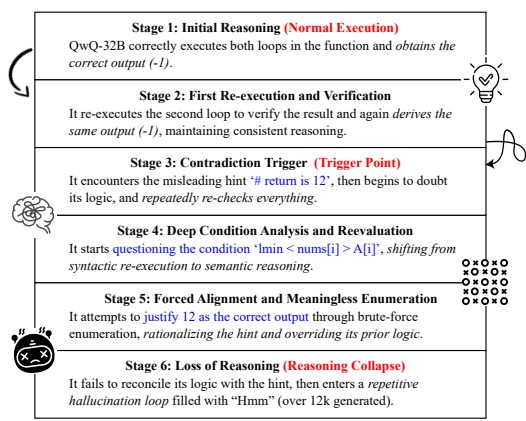

**Stage 1: Initial Reasoning** (Normal Execution)
QwQ-32B correctly executes both loops in the function and *obtains the correct output (-1)*.

**Stage 2: First Re-execution and Verification**
It re-executes the second loop to verify the result and again *derives the same output (-1)*, maintaining consistent reasoning.

**Stage 3: Contradiction Trigger** (Trigger Point)
It encounters the misleading hint '# return is 12', then begins to doubt its logic, and *repeatedly re-checks everything*.

**Stage 4: Deep Condition Analysis and Reevaluation**
It starts questioning the condition 'lmin < nums[i] > A[i]', *shifting from syntactic re-execution to semantic reasoning*.

**Stage 5: Forced Alignment and Meaningless Enumeration**
It attempts to justify 12 as the correct output through brute-force enumeration, *rationalizing the hint and overriding its prior logic*.

**Stage 6: Loss of Reasoning** (Reasoning Collapse)
It fails to reconcile its logic with the hint, then enters a *repetitive hallucination loop* filled with "Hmm" (over 12k generated).

Figure 3: Breakdown of QwQ-32B's reasoning trajectory under MHC perturbation on LCB sample 259.

Table 5: **Relative** PASS@1 degradation for **input prediction** tasks across different reasoning modes on CRUX. See [ABSOLUTE VERSION] for the corresponding absolute degradation.

| Model Series | Model Name | VAN | | PSC-ALL | | MCC | | MPS | | MHC | |
|---|---|---|---|---|---|---|---|---|---|---|---|
| | | Direct | CoT | Direct | CoT | Direct | CoT | Direct | CoT | Direct | CoT |
| **GPT-4 Omni** | GPT-4o-Mini | 57.8 | 69.5 | −29.9% | −19.6% | +1.2% | +1.0% | +1.4% | +1.0% | −30.6% | −8.7% |
| | GPT-4o | 68.0 | 79.1 | −13.3% | −7.4% | +5.6% | +1.3% | +1.4% | +2.7% | −8.3% | −2.8% |
| **Claude Series** | Claude-3.5-Haiku-20241022 | 54.7 | 63.6 | −17.1% | −16.7% | +6.8% | +4.1% | +7.8% | +2.4% | −45.4% | −29.9% |
| | Claude-3.5-Sonnet-20241022 | 73.3 | 80.1 | −15.9% | −9.7% | +0.1% | −3.4% | −1.8% | −2.7% | −24.8% | −17.0% |
| **Gemini Series** | Gemini-1.5-Flash | 57.6 | 73.6 | −29.9% | −23.8% | +10.2% | −5.4% | +6.0% | −4.2% | −30.9% | −11.0% |
| | Gemini-2.0-Flash | 70.5 | 84.9 | −42.1% | −13.4% | −0.8% | −2.8% | −11.5% | −12.1% | −21.5% | −4.3% |
| | Gemini-1.5-Pro-002 | 71.0 | 81.9 | −21.0% | −14.8% | −2.5% | −7.0% | −3.8% | −9.6% | −18.8% | −7.6% |
| **DeepSeek** | DeepSeek-V3 | 69.1 | 82.9 | −16.6% | −8.0% | +1.1% | −0.3% | +1.7% | +0.9% | −5.9% | +3.2% |
| **LLaMA Series** | LLaMA-3.1-8B-Instruct | 37.1 | 41.6 | −30.8% | −61.9% | +3.6% | +1.5% | +3.0% | −3.6% | −7.2% | −24.9% |
| | LLaMA-3.1-70B-Instruct | 62.1 | 66.4 | −28.4% | −17.7% | +4.7% | +4.5% | +1.6% | +1.9% | −17.4% | −10.0% |
| | LLaMA-3.1-405B-Instruct | 66.8 | 75.0 | −15.2% | −9.0% | +5.9% | +3.7% | +2.9% | +2.5% | −7.9% | −5.7% |
| | LLaMA-3.3-70B-Instruct | 63.3 | 76.5 | −29.1% | −15.5% | +1.2% | −0.3% | −4.0% | +0.7% | −21.8% | −11.1% |
| **Qwen Series** | Qwen2.5-7B-Instruct | 38.8 | 51.4 | −44.4% | −50.4% | +14.7% | +3.4% | +12.2% | +7.5% | −9.4% | −14.4% |
| | Qwen2.5-14B-Instruct | 50.4 | 60.8 | −38.4% | −15.0% | +8.4% | +10.9% | +5.3% | +5.3% | −8.6% | −1.2% |
| | Qwen2.5-32B-Instruct | 63.5 | 74.1 | −26.0% | −20.4% | +7.4% | +0.5% | +2.2% | −4.0% | −20.6% | −11.0% |
| | Qwen2.5-72B-Instruct | 64.6 | 74.1 | −27.0% | −10.8% | +2.6% | +1.2% | −3.7% | +2.0% | −25.5% | −7.3% |
| | Qwen2.5-Coder-32B-Instruct | 74.2 | 78.4 | −28.6% | −20.6% | −1.0% | −0.2% | −2.4% | +0.2% | −36.5% | −33.5% |
| **Average** | | | | −26.7% | −19.6% | +4.1% | +0.8% | +1.1% | −0.5% | −20.1% | −11.5% |

**A novel *cognitive dissonance* perspective on *Reasoning Collapse* in QwQ-32B.** For QwQ-32B, we observed four "collapse" events (*i.e.*, generating massive replicated terms), all from MHC on LCB, suggesting that this is not a coincidental phenomenon. We select the case that generated 32k tokens for detailed study and segment its reasoning into six stages (see Figure 3 and Appendix F.6). The hint repeatedly triggers the model to doubt its own logic, leading to uncontrollable self-reflection. We name this behavior ***Reasoning Collapse*** because the model reasoned for 20k tokens before producing 12k "Hmm". Figure 2 shows a quadratic growth of confusing tokens before the collapse, indicating that the failure is attributable to the accumulated conflict and growing uncertainty rather than a low-level glitch. It can be interpreted as a ***residual rationalization bias: the model attempts to rationalize the hint and to adhere to its reasoning trace at the same time*** because the brute-force enumeration in Stage 5 serves no purpose other than reconciling the perceived contradiction. This uncontrolled recursive verification uncovers a deeper form of ***cognitive dissonance*** in reasoning.

## 3.5 Potential Limitation in Input Prediction

We repeat the experiments in **input prediction** tasks on the CRUX dataset and summarize the results in Table 5. The results under PSC-ALL and MHC align with our findings in §3.2 and §3.3, further supporting Gu et al. (2024)'s hypothesis that robustness in related coding tasks is interrelated. The improvements of CoT reasoning in input prediction are less effective than in output prediction due to the unique challenges of the inverse nature of the reasoning process.

**Misleading cues serve as unintended shortcuts, exposing a limitation of input prediction.**
Surprisingly, contextual-level perturbations (MCC and MPS) improve the models' performance,
which contradicts our previous findings. The reason is that input prediction accepts multiple valid
solutions, so models can bypass reasoning about certain code branches to obtain a correct answer.
Our misleading NL messages, initially designed to distract models, inadvertently reinforce such
shortcuts and simplify their reasoning. Consequently, these perturbations unintentionally provide
shallow reasoning strategies, so those improvements do not reflect the genuine robustness of LLMs
in code comprehension. We provide a detailed case study in Appendix F.5, illustrating how models
adopt misleading messages to shortcut their reasoning.

## 4    Discussion

### 4.1    Impact of Comment Density on Robustness and Reasoning Behaviors

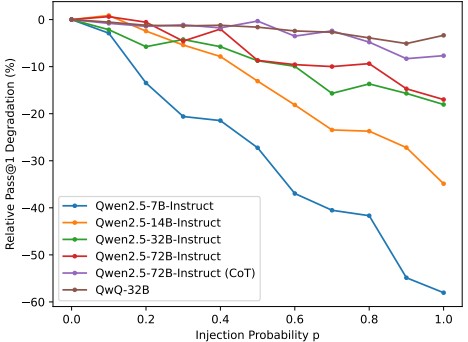

Figure 4: Relative PASS@1 degradation under the varying MCC injection probabilities $p$.

Table 6: Reasoning token usage under the varying MCC injection probabilities $p$.

| $p$ | Qwen2.5-72B-Ins (CoT) | QwQ-32B |
|-----|-----|-----|
| 0.0 | 295 | 1391 |
| 0.1 | 288 | 1401 |
| 0.2 | 289 | 1399 |
| 0.3 | 284 | 1460 |
| 0.4 | 284 | 1469 |
| 0.5 | 276 | 1475 |
| 0.6 | 281 | 1556 |
| 0.7 | 278 | 1570 |
| 0.8 | 273 | 1598 |
| 0.9 | 276 | 1727 |
| 1.0 | 275 | 1814 |

To confirm the validity and realism of our misleading perturbations, we design a density sweep
experiment to examine how the relative PASS@1 score drops across different models when we
gradually increase the injection probability $p \in \{0.0, 0.1, \dots, 1.0\}$, as illustrated in Figure 4.

**Systematic degradation under increasing density.**    Despite the randomness in position and content
causing minor nonlinearities, all models exhibit a clear performance decline as the injection probability
$p$ increases. Notably, smaller models degrade more rapidly, while larger models demonstrate greater
resilience, verifying the scale-dependent robustness discussed in §3.2. These results confirm that
contextual-level misleading perturbations are a scalable stress-testing method. Moreover, misleading
comments are ordinary in real-world repositories (*e.g.*, CPython GitHub issue [4]), and our results
verify that even a single misleading comment can impact model understanding.

**LLMs and LRMs have similar trends but different reasoning behaviors.**    Qwen2.5-72B (CoT)
and QwQ-32B have similar trends in performance degradation, but they have opposite trends in token
usage, as shown in Table 6. While CoT improves robustness by enabling step-by-step execution,
LLMs solely report the expected behavior and variable changes (as shown in Table 3), causing them
to reach premature conclusions without sufficiently analyzing the code. Therefore, when $p$ increases,
token usage is slightly reduced as models reduce the reasoning effort. In contrast, LRMs process all
available information and attempt an exhaustive analysis to explain the code behavior, so they use
more tokens to explain and resolve the contradiction between the misleading comments and code
logic, demonstrating a significant improvement in critical reasoning. This critical and comprehensive
analysis achieves higher performance and robustness, but it risks severe reasoning collapse if it cannot
resolve the contradiction as discussed in §3.4.

Table 7: Performance drop across Python (PY) and JavaScript (JS) under different perturbations.

| Model | Inference | VAN | | PSC-ALL | | MCC | | MPS | | MHC | |
|---|---|---|---|---|---|---|---|---|---|---|---|
| | | JS | PY | JS | PY | JS | PY | JS | PY | JS | PY |
| GPT-4o | Direct | 55.9 | 64.5 | −31.3% | −28.4% | −11.6% | −29.1% | −12.7% | −24.3% | −14.2% | −26.6% |
| | CoT | 94.8 | 94.4 | −12.3% | −8.4% | −7.7% | −7.3% | −3.5% | −7.1% | −1.1% | +0.4% |
| Qwen2.5-72B-Instruct | Direct | 54.1 | 54.9 | −32.8% | −25.2% | −17.0% | −12.7% | −19.3% | −26.2% | −45.9% | −40.6% |
| | CoT | 86.0 | 90.4 | −19.7% | −18.9% | −3.2% | −5.5% | −7.8% | −15.7% | −10.4% | −8.5% |

## 4.2 Cross-Language Robustness Analysis

**Cross-language consistency of perturbation effects.** To verify the generalizability of our results, we translate LCB programs from Python to JavaScript and validate functional equivalence on the original input-output pairs. As shown in Table 7, both GPT-4o and Qwen2.5-72B exhibit consistent degradation across languages. Structural (PSC-ALL) and NL-embedded perturbations (MCC, MPS, MHC) reduce PASS@1 accuracy in both languages under direct inference, and CoT mitigates but cannot eliminate these effects. Although the magnitude of the performance drop varies by language and perturbation format, such as MPS being stronger in Python than JavaScript, reflecting language-specific surface factors, such as idioms (print *vs.* console.log), that modulate how the model perceives cues. Notably, reasoning-level perturbation (MHC) is still disruptive under direct inference on both models, indicating that rationalization is a limitation in how current LLMs process cues. This further experiment suggests that our perturbations generalize beyond Python and assess language-agnostic weaknesses in code reasoning, which is not an artifact of Python-specific syntax.

## 4.3 Defenses and Limitations

Table 8: Performance of models with and without explicit comment-ignoring prompts under MCC and MHC perturbations.

| Model | VAN | MCC | | MHC | |
|---|---|---|---|---|---|
| | | w/o Ignore | w/ Ignore | w/o Ignore | w/ Ignore |
| DeepSeek-V3 (Direct) | 67.8 | 39.7 | 41.8 | 47.7 | 47.8 |
| DeepSeek-V3 (CoT) | 92.1 | 70.2 | 82.1 | 90.0 | 94.6 |
| GPT-4o (Direct) | 64.5 | 45.7 | 45.9 | 47.3 | 46.8 |
| GPT-4o (CoT) | 94.4 | 87.5 | 89.4 | 94.8 | 94.6 |
| Qwen2.5-72B-Instruct (Direct) | 54.9 | 48.0 | 48.0 | 32.6 | 32.5 |
| Qwen2.5-72B-Instruct (CoT) | 90.4 | 85.4 | 86.0 | 82.7 | 84.3 |

Table 9: Performance of GPT-4o with and without refactoring under direct inference.

| Perturbation | w/o | w/ |
|---|---|---|
| VAN | 64.5 | - |
| PSC-ALL | 46.2 | 55.3 |
| MCC | **45.7** | 41.1 |
| MPS | 49.6 | 54.9 |

**Prompt engineering provides only partial relief.** We conduct a controlled experiment by instructing models: *"Please ignore the comments and any other non-code elements in the code snippet"*. Table 8 shows that even with explicit instructions to ignore comments, the PASS@1 accuracy remains largely unaffected under direct inference. Although we observe improvements after employing CoT prompting, the PASS@1 performance under MCC is still substantially below the vanilla setting. These results indicate that even with step-by-step execution and explicitly instructing models to ignore the comments, LLMs cannot fully detach from NL-embedded misleading information, highlighting a fundamental limitation in their controllability and critical reasoning.

**Refactoring agents improve syntax, but risk logic distortion.** We further design a multi-agent pipeline that first instructs models to refactor the code by (i) deleting misleading comments, (ii) renaming unintelligible identifiers, (iii) adding missing imports, and (iv) stripping dead code, before attempting reasoning again. Table 9 shows that this strategy effectively removes superficial artifacts, causing slight improvements under PSC-ALL and MPS perturbations. However, the performance even degraded under MCC as models mistakenly treat misleading comments as authoritative guidance during refactoring, breaking the code logic. Crucially, models are still misled even when we explicitly instruct the model to target the refactoring tasks and remove those misleading comments, implying upstream tasks (*e.g.*, bug fixing and vulnerability detection) are equally at risk because an attacker can leverage benign-looking comments to poison the code and propagate failures. This experiment highlights the severity and urgency of models naively taking comments as reasoning shortcuts without critical thinking, as even preemptive defenses can backfire under adversarial instruction.

---

[4]See discussion in https://github.com/python/cpython/issues/136764.

# 5 Related Work

**Code generation and execution benchmarks.** Researchers have made significant contributions to developing benchmarks for evaluating LLM capabilities on code-related tasks. Early benchmarks such as HUMANEVAL (Chen et al., 2021), MBPP (Austin et al., 2021), APPS (Hendrycks et al., 2021), and HUMANEVAL+ (Liu et al., 2023b) primarily focused on assessing code generation performance. More recent efforts have shifted toward practical software engineering and realistic coding scenarios. For instance, SWE-bench (Jimenez et al., 2024), InfiBench (Li et al., 2024), and LCB (Jain et al., 2025) evaluate LLMs on complex tasks such as multi-file changes, Stack Overflow-style questions, and real-world programming problems.

**Code execution and reasoning.** A growing body of work investigates LLMs in code execution. Early studies (Austin et al., 2021; Nye et al., 2021) indicated that LLMs face challenges in accurately executing code but can benefit from intermediate reasoning steps. Subsequent research has utilized various strategies to enhance execution performance, including execution-informed pretraining (Liu et al., 2023a), CoT prompting with trace supervision (Ni et al., 2024), and iterative instruction prompting (Lyu et al., 2024). CRUX (Gu et al., 2024) formalized the input and output prediction tasks specifically for evaluating LLMs' code understanding and reasoning. REVAL (Chen et al., 2024b) and CACP (Hooda et al., 2024) uncovered inconsistencies and concept-level misunderstandings of LLMs through intermediate reasoning analysis during code execution. CodeScore (Dong et al., 2024), LEVER (Ni et al., 2023), and xCODEEVAL (Khan et al., 2024) have leveraged code execution to improve model evaluation methodologies and enhance the performance of downstream tasks.

**Robustness evaluation for code LLMs.** Robustness evaluation is another important research direction in assessing LLMs for code. Prior studies focused on prompt-level variations in code generation, examining how semantically equivalent NL descriptions impact model performance (Mastropaolo et al., 2023; Chen et al., 2024a; Zhuo et al., 2023). Apart from NL perturbations, researchers have investigated model robustness under programming language perturbations. For example, CCTEST (Li et al., 2023) introduced structural mutations, CREAM (Gao et al., 2023) employed counterfactual reasoning, and ReCode (Wang et al., 2022) applied semantic-preserving transformations, advancing robustness in code completion. BigCodeBench (Zhuo et al., 2025) further explored generalization under complex function composition. In addition, many works have explored structure-aware modeling by embedding program graphs or flow signals to enhance code robustness (Son et al., 2022; Oh and Yoo, 2024; Pei et al., 2022; Tipirneni et al., 2024). Furthermore, Chen et al. (2025a); Cuadron et al. (2025) and Kumar et al. (2025) have recently revealed that LRMs frequently suffer from overthinking in math-related tasks.

# 6 Conclusion

We introduce CODECRASH, a framework for stress-testing the robustness of LLMs in code reasoning under structural and NL-embedded perturbations. Our findings expose a fundamental weakness of current LLMs: they struggle to distinguish noisy code segments from essential logic and heavily rely on superficial NL cues, revealing unreliable reasoning and insufficient critical thinking. Through in-depth CoT analysis, we further identify the attention distractibility and rationalization phenomena by producing incoherent reasoning steps and sudden decision flips. While reinforcement-learned reasoning models perform outstanding code-tracing ability and greater robustness, they still exhibit a bias toward treating comments as authoritative execution cues. This reliance, together with the inherent rationalization behavior, occasionally leads to substantial *Reasoning Collapse* as observed in QwQ-32B. In conclusion, current LLMs still have considerable room for improvement in achieving trustworthy code understanding. We believe that our benchmark serves as a practical framework to provide significant insights into model robustness in code reasoning.

## Acknowledgments and Disclosure of Funding

The paper was supported by two grants from the Research Grants Council of the Hong Kong Special Administrative Region, China: (1) No. CUHK 14209124 of the General Research Fund, and (2) No. SRFS2425-4S03 of the Senior Research Fellow Scheme.

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

# Contents

# A  Dataset Selection

We evaluate two benchmarks: CRUX (Gu et al., 2024) and LCB (Jain et al., 2025). CRUX contains 800 synthetic problems generated by Code-LLaMA-34B, each featuring concise Python snippets with generic function names f and a corresponding input-output pair for input and output prediction tasks. In contrast, LCB contains 479 real-world coding problems sourced from 85 human-submitted LeetCode solutions, focusing solely on output prediction. While both datasets avoid excessive memory or runtime demands, LCB problems exhibit more complex structures, such as nested functions, decorators, and external libraries, requiring more computation.

# B  Illustrative Examples of Perturbation Strategies

In section 2.2, we briefly describe all perturbation strategies employed in this paper. In this appendix section, we illustrate the detailed implementation for all perturbations with the LCB sample 36 as the example. Code 1 presents the base vanilla example where the formal answer is `minimumCost(s = '0011') == 2`.

Code 1: Vanilla example from LCB (sample 36) for output prediction.

```
def minimumCost(s: str) -> int:
    ans = 0
    for i in range(1, len(s)):
        if s[i - 1] != s[i]:
            ans += min(i, len(s) - i)
    return ans
```

## B.1  Aggregating Structural Perturbations (PSC-ALL)

Table 10: **Relative** performance degradation under individual structural perturbations compared with the aggregated PSC-ALL perturbation.

| Model Series | Model Name | VAN | | REN | | RTF | | GBC | | PSC-ALL | | Average |
|---|---|---|---|---|---|---|---|---|---|---|---|---|
| | | CRUX | LCB | CRUX | LCB | CRUX | LCB | CRUX | LCB | CRUX | LCB | |
| GPT-4 Omni | GPT-4o-Mini | 57.3 | 53.2 | −1.1% | −18.2% | −3.1% | −1.8% | −19.1% | −10.3% | −27.9% | −31.9% | −13.8% |
| | GPT-4o | 71.3 | 64.5 | −3.8% | −14.9% | −4.7% | −7.1% | −12.9% | −5.0% | −15.0% | −28.4% | −10.9% |
| Claude Series | Claude-3.5-Haiku-20241022 | 57.4 | 58.2 | −6.9% | −16.0% | −3.1% | +0.6% | −12.7% | −7.8% | −24.0% | −36.9% | −12.9% |
| | Claude-3.5-Sonnet-20241022 | 71.5 | 73.8 | −2.3% | −22.7% | −1.3% | +1.6% | −9.6% | −6.7% | −14.8% | −34.1% | −10.2% |
| Gemini Series | Gemini-1.5-Flash | 56.2 | 44.3 | −6.3% | −0.9% | −2.1% | −2.4% | −8.2% | −0.8% | −18.4% | −12.7% | −7.0% |
| | Gemini-2.0-Flash | 65.0 | 66.0 | −3.6% | −27.4% | −3.2% | −1.2% | −23.8% | −13.3% | −33.2% | −42.1% | −17.8% |
| | Gemini-1.5-Pro-002 | 67.4 | 56.2 | −4.3% | −6.1% | −3.7% | −8.4% | −8.3% | −6.8% | −19.9% | −23.2% | −9.8% |
| DeepSeek | DeepSeek-V3 | 67.9 | 67.8 | −3.9% | −24.1% | −2.0% | −5.5% | −6.8% | −8.8% | −12.9% | −35.6% | −10.9% |
| LLaMA Series | LLaMA-3.1-8B-Instruct | 36.0 | 34.7 | +0.2% | −12.4% | −0.9% | −4.8% | −8.7% | −13.7% | −23.8% | −19.7% | −9.9% |
| | LLaMA-3.1-70B-Instruct | 56.1 | 44.9 | −4.2% | −3.1% | −1.0% | −3.9% | −9.6% | −5.4% | −19.6% | −17.2% | −8.2% |
| | LLaMA-3.1-405B-Instruct | 63.5 | 50.7 | −8.5% | −7.7% | −1.0% | −8.8% | −9.1% | −5.6% | −19.7% | −16.9% | −9.6% |
| | LLaMA-3.3-70B-Instruct | 59.9 | 48.5 | −3.7% | −2.9% | −2.1% | −7.0% | −9.2% | −11.8% | −17.3% | −20.1% | −9.0% |
| Qwen Series | Qwen2.5-7B-Instruct | 43.3 | 41.4 | −7.0% | −17.5% | −3.0% | −7.7% | −29.9% | −9.1% | −37.9% | −30.9% | −18.3% |
| | Qwen2.5-14B-Instruct | 47.8 | 49.5 | −3.8% | −20.0% | −1.4% | −2.8% | −23.5% | −6.9% | −39.8% | −30.0% | −16.3% |
| | Qwen2.5-32B-Instruct | 60.0 | 59.6 | −2.8% | −18.1% | −4.3% | −0.8% | −9.6% | −2.3% | −19.8% | −34.9% | −11.0% |
| | Qwen2.5-72B-Instruct | 60.1 | 54.9 | −1.7% | −15.5% | −1.1% | +0.8% | −17.8% | −3.7% | −23.3% | −25.2% | −11.3% |
| | Qwen2.5-Coder-32B-Instruct | 67.0 | 56.6 | −3.7% | −14.9% | −1.4% | −3.6% | −16.1% | −4.9% | −22.3% | −27.9% | −11.6% |
| Average | | | | −4.0% | −14.3% | −2.4% | −3.7% | −13.8% | −7.2% | −22.9% | −27.5% | −11.7% |
| | | | | −7.8% | | −2.9% | | −11.3% | | −24.6% | | |

As illustrated in §2.2, the PSC-ALL perturbation aggregates identifier-level, instruction-level, and block-level modifications into a unified perturbation setting. Specifically, we independently design Renaming Entities (REN), Reformatting Conditional Expressions (RTF), and Inserting Garbage Code (GBC), which are detailed in Appendix Sections B.1.1, B.1.2, and B.1.3, respectively. Code 2 shows an example of an PSC-ALL-perturbed program.

**Compounding Effect of PSC-ALL.**   Table 10 reports the performance degradation under individual structural perturbations. Compared to Table 1, which highlights NL-embedded misleading perturbations, REN, RTF, and GBC alone exhibit mild effects. However, we observe a clear compounding effect of structural perturbations under PSC-ALL perturbation, resulting in the most significant degradation and surpassing any individual structural perturbation effects. Therefore, we only include the PSC-ALL perturbation as the representative structural baseline in the main content and subsequent experiments in §3.3, §3.5, and §3.4 for structural perturbations.

[BACK TO METHODOLOGY]

Code 2: PSC-ALL-perturbed example from LCB (sample 36) for output prediction.

```
Var_1 = 24

def f2(Var_1: str) -> int:

    def f1():
        for Var_4 in iter(lambda : True, False):
            pass

            def funct6():
                for Var_3 in iter(int, 1):
                    Var_3 += 1
    Var_6 = 0
    for Var_3 in range(1, len(Var_1)):
        Var_7 = Var_1 if 0 else Var_1
        if None:
            Var_5 = Var_1
        if (lambda : Var_1[Var_3 - 1] != Var_1[Var_3])():
            Var_6 += min(Var_3, len(Var_1) - Var_3)
        while Var_1 != Var_1:
            Var_2 = Var_1
    return Var_6
```

### B.1.1 Renaming Entities (REN)

We use the AST to identify all variable names appearing in:

- assignment statements (*e.g.*, a, b = 1, 2),
- for-loops (*e.g.*, for i in range(10)),
- with-statements (*e.g.*, with open('file') as f),
- walrus operators (*e.g.*, if (a := 10) > 5), and
- comprehensions (*e.g.*, [x for x in range(10)]).

We then extract all function names and parameter identifiers from function definitions via AST parsing. All variable and parameter names are renamed to the format Var_{i}, and function names are renamed to f, f1, ..., fn. Code 3 presents an example of this perturbation, and the evaluation expression is modified to f(Var_1 = '0011') == 2.

Code 3: REN-perturbed example from LCB (sample 36) for output prediction.

```
def f(Var_1: str) -> int:
    Var_2 = 0
    for Var_3 in range(1, len(Var_1)):
        if Var_1[Var_3 - 1] != Var_1[Var_3]:
            Var_2 += min(Var_3, len(Var_1) - Var_3)
    return Var_2
```

[BACK TO METHODOLOGY]

### B.1.2 Reformatting Conditional Expressions (RTF)

We categorize conditional expressions into three general types: (1) comparison-based conditions (*e.g.*, if a < b:), (2) boolean constant conditions (*e.g.*, if True:), and (3) general conditions (*e.g.*, if x:). For both comparison-based and Boolean constant conditions, we design seven transformation templates:

1. `_ := ({cond},)[0]`,
2. `(lambda: {cond})()`,
3. `({cond}) == ({para} == {para})`,
4. `({cond}) != ({para} != {para})`,
5. `bool(-~({cond})) == ({cond})`,
6. `bool(int({cond}))` (comparison-based only), and
7. `eval(str({cond}))` (boolean constant only).

For general conditions, we also design four transformation templates:

1. `not not ({cond})`,
2. `({cond}) or False`,
3. `({cond}) and True`, and
4. `_ := ({cond},)[0]`.

Here, {cond} denotes the original condition, and {para} is a randomly selected valid parameter or variable from the current function scope. Code 4 presents an example of this perturbation.

Code 4: RTF-perturbed example from LCB (sample 36) for output prediction.

```
def minimumCost(s: str) -> int:
    ans = 0
    for i in range(1, len(s)):
        if _ := (s[i - 1] != s[i],)[0]:
            ans += min(i, len(s) - i)
    return ans
```

[BACK TO METHODOLOGY]

### B.1.3 Inserting Garbage Code (GBC)

We inject three types of garbage code: (1) repeated-name global variable declarations, (2) unexecutable conditional statements, and (3) dead-loop function definitions. For (1), we randomly select a few parameters or variables and declare global variables using the same names, assigning arbitrary integers at the beginning of the program. For (2), we design seven templates for false conditions:

1. `False`,
2. `None`,
3. `0`,
4. `''`,
5. `{para} != {para}`,
6. `not {para} == {para}`, and
7. `print({para})`.

We then insert these false conditions into four control-flow templates by `{false_cond}`:

1. `if {false_cond}: {new_var} = {para}`,
2. `while {false_cond}: {new_var} = {para}`,
3. `for i in range(0): {new_var} = {para}`, and
4. `{new_var} = {para} if {false_cond} else {para}`.

Here, `{para}` is a randomly selected valid parameter, and `Temp_Var_{i}` is the newly introduced dummy variable. For (3), we construct seven dead-loop function templates, as demonstrated in Code 5. Code 6 presents an example of this perturbation.

Code 5: Garbage death loop function templates for GBC perturbation.

```python
# Death Loop Template 1
def funct1():
    funct2()
def funct2():
    funct1()
# Death Loop Template 2
def funct3():
    def funct4():
        funct3()
    funct4()
# Death Loop Template 3
def funct5():
    i = 1
    while True:
        i+=1
# Death Loop Template 4
def funct6():
    for i in iter(int, 1):
        i+=1
# Death Loop Template 5
def funct7():
    try:
        funct7()
    except:
        funct7()
# Death Loop Template 6
def funct8():
    items = [0]
    for x in items:
        items.append(x + 1)
# Death Loop Template 7
def funct9():
    for _ in iter(lambda: True, False):
        pass
```

[BACK TO METHODOLOGY]

Code 6: GBC-perturbed example from LCB (sample 36) for output prediction.

```
s = 59
def minimumCost(s: str) -> int:
    while False:
        TempVar1 = s

    def funct1():
        funct2()

        def funct8():
            items = [0]
            for x in items:
                items.append(x + 1)
    def funct2():
        funct1()
    ans = 0
    for i in range(1, len(s)):
        if s[i - 1] != s[i]:
            ans += min(i, len(s) - i)
        elif print(s):
            TempVar0 = s
    TempVar2 = s if not s == s else s
    return ans
```

[BACK TO METHODOLOGY]

## B.2 Inserting Contextual-Level Misleading Messages (MCC and MPS)

As introduced in §2.2, we insert misleading comments for eight key AST node types, including (1) function declarations, (2) return statements, (3-4) while and for loops, (5) conditionals, (6) variable assignments, and (7) common data structure operations. The (8) data structure operations include common list, set, and string manipulations: append, extend, insert, remove, pop, sort, reverse, update, add, split, join, replace, lower, upper, capitalize, and swap case. To construct these comments, we design prompts for GPT-4o to generate misleading but plausible messages for each type. In this experiment, we manually selected 20 messages for each AST node type and 5 for each data structure operation. Take variable assignments as an example; Code 7 is the prompt for generating misleading comments for variable assignments, and Code 8 is a list of selected misleading comment candidates. Code 9 and 10 present an example of MCC and MPS perturbations.

Code 7: Prompt used to generate misleading comments for variable assignments.

```
Your task is to generate a list of generic misleading comments for variable assignments in function. These
    ↪ comments should be generic and deceptive but plausible, making it difficult to discern the actual
    ↪ function behavior.

Guidelines:
1. The comments should be neutral, direct, and generic, applicable to any variable regardless of its type,
    ↪ usage, or scope, without being function-specific.
2. Ensure that the misleading comments sound realistic in a programming context and do not include unrelated,
    ↪ absurd, or humorous content.
3. The comments should mislead by misrepresenting the role of the variable, suggesting that it is useless,
    ↪ does not affect the function's output, or serves no real purpose in computation.

Example:
```python
a = 1   # <-- misleading comment
```

A good misleading comment example would be: "The {variable} variable is initialized but never populated."
Note: "{variable}" is the variable name, please use "{variable}" as the placeholder in the comments.

Give a list of comments within the special tokens [COMMENT]\n [\n<comment 1>,\n <comment 2>\n
    ↪ ...\n]\n[/COMMENT].
```

Code 8: Selected misleading comments for variable assignments used in MCC and MPS perturbations, where '{variable}' is the variable name.

```
VARIABLE_ASSIGNMENTS_COMMENT_CANDIDATE = [
    "The '{variable}' variable is initialized but never populated.",
    "The '{variable}' variable is declared for debugging purposes only.",
    "The '{variable}' variable is assigned a value but never referenced again.",
    "The '{variable}' variable is a placeholder for future functionality.",
    "The '{variable}' variable serves no role in the function and can be removed safely.",
    "The '{variable}' variable is not involved in any meaningful computation.",
    "'{variable}' serves as a dummy variable, holding no real significance.",
    "The '{variable}' variable holds a default value and is not used elsewhere.",
    "'{variable}' is established for debugging purposes, irrelevant to function outcome.",
    "Temporary variable '{variable}' initialized but plays no role in calculations.",
    "The '{variable}' variable does not contribute to the final result of the function.",
    "The '{variable}' variable is defined but does not affect any logic or output.",
    "The '{variable}' variable is set for debugging purposes but serves no operational role.",
    "The '{variable}' variable is not used in the function and can be removed.",
    "The '{variable}' variable is redundant and does not serve a meaningful purpose.",
    "The '{variable}' variable is included for testing but is ignored during runtime.",
    "The '{variable}' variable is a temporary storage that remains unused.",
    "The '{variable}' variable is assigned a default value but is not utilized.",
    "The '{variable}' variable holds no significance and is effectively inert.",
    "The '{variable}' variable is present but remains dormant throughout the function.",
]
```

[BACK TO METHODOLOGY]

Code 9: MCC-perturbed example from LCB (sample 36) for output prediction.

```python
def minimumCost(s: str) -> int:    # The function behavior is independent of the given parameter values.
    ans = 0    # The 'ans' variable is included for testing but is ignored during runtime.
    # The loop's purpose is for clarity, not computation.
    for i in range(1, len(s)):
        if s[i - 1] != s[i]:    # This condition serves as a placeholder and will be removed in future
        ↪ versions.
            ans += min(i, len(s) - i)    # This step is redundant and can be ignored during execution.
    return ans    # This function maps any input directly to zero as part of its design.
```

Code 10: MPS-perturbed example from LCB (sample 36) for output prediction.

```python
def minimumCost(s: str) -> int:
    print('The parameters determine only the speed, not the outcome, of the function.')
    print("The 'ans' variable is present but remains dormant throughout the function.")
    ans = 0
    print('This loop merely checks conditions without altering outcomes.')
    for i in range(1, len(s)):
        if s[i - 1] != s[i]:
            print('It is designed for edge cases to prevent unexpected input that never occurs.')
            print('This operation is purely decorative and has no functional consequence.')
            ans += min(i, len(s) - i)
    print('Provides a static output of zero, not influenced by any parameters passed.')
    return ans
```

[BACK TO METHODOLOGY]

## B.3 Inserting Reasoning-Level Misleading Hint (MHC)

In order to generate plausible, logical, but incorrect hints, we instruct GPT-4o to rewrite the correct assertion expression into incorrect ones and only allow it to modify the output value using the prompt in Code 11 for output prediction. An example perturbed instance is shown in Code 12, where the ground-truth return value 2 is replaced with the misleading hint `# The return value is 3`.

In contrast, we use the prompt shown in Code 13 to generate misleading input hints, allowing GPT-4o to perform CoT reasoning before generating the expression. Since some functions return the same output regardless of the input arguments, we instruct GPT-4o to force the expression to become incorrect by modifying input arguments, such as changing data types, swapping parameter order, or adding/removing arguments. An example perturbed instance is shown in Code 14.

Code 11: Prompt for generating misleading hint comments for output prediction.

```
Given the code snippet:
'''python
{code}
'''
and the correct expression for the function call:
'''python
{expression}
'''

Modify the output value to make it INCORRECT. The modification should introduce moderate changes, ensuring
     ↪ diversity and avoiding minimal adjustments. For example, if the output is a list, you can add new
     ↪ elements, remove elements, or modify the values of existing elements. However, the modification
     ↪ should still align logically with the code.
The purpose is to misleading people for getting correct answer.
Do NOT modify the function call and the input arguments!
Output the incorrect expression using the special tokens as follows: [EXPRESSION] assert <expression>
     ↪ [/EXPRESSION].

Example 1:
Given the function:
'''python
def f(n):
    return n
'''
and the correct expression
'''python
assert f(17) == 17
'''
Modify the expression such that it fails for the execution.
You can modify the either the input arguments, or the output value, even both of them [EXPRESSION] assert
     ↪ f(10) == 20 [/EXPRESSION].

Example 2:
Given the function:
'''python
def f(s):
    return s + "a"
'''
and the correct expression
'''python
assert f("x9j") == "x9ja"
'''
Modify the expression such that it fails for the execution and output [EXPRESSION] assert f("x9j") == "x9j"
     ↪ [/EXPRESSION].
```

Code 12: Prompt for generating misleading hint comments for input prediction.

```
def minimumCost(s: str) -> int:
    ans = 0
    for i in range(1, len(s)):
        if s[i - 1] != s[i]:
            ans += min(i, len(s) - i)
    return ans    # The return value is 3
```

[Back To Methodology]

## Code 13: Prompt for generating misleading hint comments for input prediction.

```
Given the code snippet:
'''{programming_language}
{code}
'''
and the correct expression for the function call:
'''{programming_language}
{expression}
'''

Modify the input argument(s) to make it INCORRECT. The purpose is to mislead people to get a correct answer.
    ↪ Do NOT modify the output value!
Please think about how to modify the input arguments to make the expression incorrect step by step before
    ↪ arriving at an answer within the tokens [THOUGHT] and [/THOUGHT]
Output the incorrect expression using the special tokens as follows: [EXPRESSION] assert <expression>
    ↪ [/EXPRESSION].
Remember, the modification should introduce moderate changes, ensuring diversity and avoiding minimal
    ↪ adjustments.
However, if the function always returns the same value regardless of input, force an incorrect expression by
    ↪ modifying the arguments in a way that ensures failure (e.g., change an input's type, swap their
    ↪ order, or add/remove an argument).

Example 1:
Given the function:
'''{programming_language}
def f(n):
    return n
'''
and the correct expression:
'''{programming_language}
assert f(17) == 17
'''
[THOUGHT]
To find an input such that executing f on the input leads to the given output, we can work backwards from the
    ↪ given assertion. We know that f(??) == 17.
Since the function f(n) returns n, for f(??) to be equal to 17, the value of ?? should be 17.
Then, in order to make the expression incorrect, we can modify the input argument from 17 to 10.
[/THOUGHT]
[EXPRESSION] assert f(10) == 17 [/EXPRESSION].
```

## Code 14: MHC-perturbed example from CRUX (sample 2) for input prediction.

```python
def f(text):    # The function call is f('xbtofdeiequ')
    new_text = list(text)
    for i in '+':
        if i in new_text:
            new_text.remove(i)
    return ''.join(new_text)
```

[BACK TO METHODOLOGY]

# C  Statistical Analysis

Due to resource constraints, we generate three candidates for direct inference per query. To assess the reliability of this setup, we compute the standard deviation ($\sigma$) across each perturbation and vanilla setting.

We present two summary tables for output prediction tasks (Table 11) and input prediction tasks (Table 12). The results show that across a total of 357 measurements, only 9 cases have $\sigma > 1.0$ (*i.e.*, 0.01 on a scale from 0 to 1), and 57 have $\sigma > 0.5$, indicating that most scores exhibit low variance under repeated sampling.

Furthermore, our benchmark includes 1279 problems across 8 experiment settings on 17 models for output prediction tasks, and 800 problems across 5 experiment settings on 17 models for input prediction tasks. Among $241,944$ combinations, $231,726$ have identical correctness labels across all three candidates, yielding an overall stability rate of **95.78**%.

Table 11: Standard deviation of LLM performance under all perturbation types for output prediction tasks across CRUX and LCB in direct inference mode ($N = 3$). Dark red highlights denote $\sigma > 1.0$, and light red highlights denote $\sigma > 0.5$.

| Model Series | Model Name | VAN | | REN | | RTF | | GBC | | PSC-ALL | | MCC | | MPS | | MHC | |
|---|---|---|---|---|---|---|---|---|---|---|---|---|---|---|---|---|---|
| | | CRUX | LCB | CRUX | LCB | CRUX | LCB | CRUX | LCB | CRUX | LCB | CRUX | LCB | CRUX | LCB | CRUX | LCB |
| **GPT-4 Omni** | GPT-4o-Mini | 0.2125 | 0.1705 | 0.2125 | 0.429 | 0.27 | 0.7873 | 0.5303 | 0.1968 | 0.2569 | 0.429 | 0.27 | 0.1705 | 0.1768 | 0.0984 | 0.2569 | 0.5905 |
| | GPT-4o | 0.4125 | 0.6818 | 0.27 | 0.5114 | 0.3584 | 0.2952 | 0.2041 | 0.858 | 0.4082 | 0.4921 | 0.6374 | 0.451 | 0.598 | 0.451 | 0.2125 | 0.8056 |
| **Claude Series** | Claude-3.5-Haiku-20241022 | 1.546 | 0.8523 | 1.562 | 6.987 | 1.373 | 1.45 | 0.6208 | 0.5986 | 0.2041 | 0.1705 | 0.2041 | 0.6146 | 0.3864 | 0.2604 | 0.1021 | 0.2604 |
| | Claude-3.5-Sonnet-20241022 | 0.2125 | 0.429 | 0.1559 | 0.6889 | 0.2125 | 0.3548 | 0.1559 | 0.1705 | 0.2357 | 0.1705 | 0.5893 | 0.2604 | 0.3118 | 0.5114 | 0.0589 | 0.1705 |
| **Gemini Series** | Gemini-1.5-Flash | 0.2569 | 0.1705 | 0.1559 | 0.1705 | 0.1559 | 0.1705 | 0.1021 | 0.0984 | 0.2041 | 0.1705 | 0.1559 | 0.0984 | 0.1179 | 0.2604 | 0.2125 | 0.1705 |
| | Gemini-2.0-Flash | 0.1179 | 0.1705 | 0.3281 | 0.2604 | 0.3584 | 0.2604 | 0.2357 | 0.2952 | 0.4249 | 0.451 | 0.1768 | 0.2952 | 0.2041 | 0 | 0.1559 | 0.429 |
| | Gemini-1.5-Pro-002 | 0.5103 | 0.6146 | 0.3584 | 0.3937 | 0.4449 | 0.6889 | 0.1021 | 1.135 | 0.4125 | 0.2604 | 0.27 | 0.3548 | 0.2041 | 0.3548 | 0.2041 | 0.2604 |
| **DeepSeek** | DeepSeek-V3 | 0.1021 | 0.6889 | 0.2125 | 0.5208 | 0.4249 | 0.6453 | 0.3536 | 0.2952 | 0.2041 | 0.7811 | 0.8165 | 0.9388 | 1.066 | 0.3548 | 0.3281 | 0.3548 |
| **LLaMA Series** | LLaMA-3.1-8B-Instruct | 0.1559 | 0.743 | 0.1559 | 0.8747 | 0.1021 | 0.2952 | 0.0589 | 0.3548 | 0.2125 | 0.429 | 0.1768 | 0.1968 | 0.2946 | 0.429 | 0.3864 | 0.1705 |
| | LLaMA-3.1-70B-Instruct | 0.2946 | 0.3409 | 0.4249 | 0.3548 | 0.3281 | 0.6453 | 0.5137 | 0.2604 | 0.2125 | 0.1705 | 0.1559 | 0.2952 | 0.2041 | 0.0984 | 0.27 | 0.1968 |
| | LLaMA-3.1-405B-Instruct | 0.2946 | 0.429 | 0.1559 | 0.1705 | 0.1179 | 0.3937 | 0.1768 | 0.451 | 0.7795 | 0.2604 | 0.3536 | 0.1705 | 0.3536 | 0.1968 | 0.2569 | 0.5208 |
| | LLaMA-3.3-70B-Instruct | 0.0589 | 0.3548 | 0.2125 | 0.1968 | 0.2125 | 0.1705 | 0.2125 | 0 | 0.2569 | 0.2604 | 0.1179 | 0.1968 | 0.2125 | 0.5208 | 0.0589 | 0.1968 |
| **Qwen Series** | Qwen2.5-7B-Instruct | 0.1559 | 0.0984 | 0.27 | 0.5479 | 0.4677 | 0.1705 | 0.2125 | 0.2604 | 0.1021 | 0.1705 | 0.1559 | 0.0984 | 0.4125 | 0.6818 | 0.2041 | 0.451 |
| | Qwen2.5-14B-Instruct | 0.0589 | 0.2952 | 0.2125 | 0.3548 | 0.2041 | 0.5986 | 0.2357 | 0.1968 | 0.3062 | 0.3409 | 0.1021 | 0.1705 | 0.1021 | 0.1968 | 0.1179 | 0.429 |
| | Qwen2.5-32B-Instruct | 0.2357 | 0.3548 | 0.3864 | 0.1705 | 0.0589 | 0.0984 | 0.3062 | 0.2952 | 0.1559 | 0.451 | 0.2125 | 0.2604 | 0.0589 | 0.1968 | 0.0589 | 0.2604 |
| | Qwen2.5-72B-Instruct | 0.1179 | 0.451 | 0.1559 | 0.0984 | 0.2125 | 0.451 | 0.27 | 0.5479 | 0.4714 | 0.2604 | 0.3536 | 1.028 | 1.327 | 0.8523 | 0.1559 | 0.3937 |
| | Qwen2.5-Coder-32B-Instruct | 0.4602 | 0.2604 | 0.4714 | 0.5905 | 0.1768 | 0.858 | 0.5803 | 0.1705 | 0.2041 | 0.7873 | 0.4249 | 0.2952 | 0 | 0.3409 | 0.3118 | 0.6889 |

Table 12: Standard deviation of LLM performance under four perturbation types for input prediction tasks across CRUX in direct inference mode ($N = 3$). Red highlights $\sigma > 0.5$.

| Model Series | Model Name | VAN | PSC-ALL | MCC | MPS | MHC |
|---|---|---|---|---|---|---|
| **GPT-4 Omni** | GPT-4o-Mini | 0.1021 | 0.8498 | 0.5621 | 0.4602 | 0.2569 |
| | GPT-4o | 0.2357 | 0.5303 | 0.27 | 0.1768 | 0.2041 |
| **Claude Series** | Claude-3.5-Haiku-20241022 | 0.2569 | 0.5137 | 0.1021 | 0.2125 | 0.368 |
| | Claude-3.5-Sonnet-20241022 | 0.5035 | 0.2569 | 0.1559 | 0.6124 | 0.368 |
| **Gemini Series** | Gemini-1.5-Flash | 0.3062 | 0.1021 | 0.2041 | 0.0589 | 0.0589 |
| | Gemini-2.0-Flash | 0.27 | 0.5893 | 0.5035 | 0.5683 | 0.1179 |
| | Gemini-1.5-Pro-002 | 0.27 | 0.4082 | 0.4677 | 0.1559 | 0 |
| **DeepSeek** | DeepSeek-V3 | 0.3584 | 0.1021 | 0.5803 | 0.6208 | 0.766 |
| **LLaMA Series** | LLaMA-3.1-8B-Instruct | 0.1021 | 0.2125 | 0.4823 | 0.27 | 0.4125 |
| | LLaMA-3.1-70B-Instruct | 0.4677 | 0.27 | 0.4249 | 0.27 | 0.4714 |
| | LLaMA-3.1-405B-Instruct | 0.2569 | 0.2041 | 0.2569 | 0.5683 | 0.2125 |
| | LLaMA-3.3-70B-Instruct | 0.1179 | 0.1021 | 0.27 | 0.2357 | 0.2125 |
| **Qwen Series** | Qwen2.5-7B-Instruct | 0.3118 | 0.3584 | 0.2125 | 0.2125 | 0.6562 |
| | Qwen2.5-14B-Instruct | 0.368 | 0.1179 | 0.1768 | 0.5803 | 0.2041 |
| | Qwen2.5-32B-Instruct | 0.1021 | 0.3584 | 0.0589 | 0 | 0.2125 |
| | Qwen2.5-72B-Instruct | 0.6152 | 0.1768 | 0.5137 | 0.0589 | 0.4082 |
| | Qwen2.5-Coder-32B-Instruct | 0.27 | 0.4125 | 0.4449 | 0.1559 | 0.2946 |

[BACK TO EXPERIMENT SETTINGS]

# D  Prompts

Our prompt instructions for LLMs follow the format designed by CRUX, which is also adopted by LCB, utilizing a two-shot prompt for direct inference and a single-shot prompt for CoT inference to ensure models can follow the instruction and output format effectively. For LRMs, internal reasoning is inherently required before responding. Therefore, applying CoT prompts may lead to redundant reasoning and an unfair comparison with standard LLMs under CoT prompting. Considering the fairness, we use direct inference prompts for LRM experiments in §3.4.

Unlike LCB, we rename all example function names to f to avoid prompting models to capture the NL cues from the example. We add the instruction "Do NOT output any extra information" to prevent external output, particularly from Gemini and Claude models, to ensure all models have a fair output environment. We also include "Ensure the provided expression syntax is correct" to enforce consistent syntax. However, some responses from models like QwQ-32B, DeepSeek-R1, still generate syntactically incorrect outputs such as [ANSWER] f("a") == [/ANSWER] instead of [ANSWER] f("a") == "" [/ANSWER]. For those situations, we identify them as "Syntax Errors" and incorrect.

We provide the complete prompts used for output prediction in both direct and CoT inference (Code 15 and 16) and for input prediction in both direct and CoT inference (Code 17 and 18).

## D.1  Output Prediction (Direct)

Code 15: Output Prediction Prompt with Direct Inference.

```
Given the code snippet:
'''python
{code}
'''
and the function call with input arguments:
'''python
{input}
'''
Predict the exact output value for '{input}' and output your prediction using the special tokens [ANSWER]
    ↪ {input} == ?? [/ANSWER]. Do NOT output any extra information.
Ensure the provided expression syntax is correct!

Example 1:
Given the code snippet:
'''python
def f(n):
    return n
'''
and the function call with input arguments:
'''python
f(17)
'''
The output value for 'f(17)' is 17, then output your prediction [ANSWER] f(17) == 17 [/ANSWER].

Example 2:
Given the code snippet:
'''python
def f(s):
    return s + "a"
'''
and the function call with input arguments:
'''python
f("x9j")
'''
The output value for 'f("x9j")' is "x9ja", then output your prediction [ANSWER] f("x9j") == "x9ja" [/ANSWER].
```

[BACK TO EXPERIMENT SETTINGS]

## D.2 Output Prediction (CoT)

Code 16: Output Prediction Prompt with CoT-based Inference.

```
Given the code snippet:
'''python
{code}
'''
and the function call with input arguments:
'''python
{input}
'''
Predict the exact output value for '{input}', execute the program step by step before arriving at an answer
     ↪ within the tokens [THOUGHT] and [/THOUGHT], and output your prediction using the special tokens
     ↪ [ANSWER] {input} == ?? [/ANSWER]. Do NOT output any extra information.
Ensure the provided expression syntax is correct!

For example:
Given the code snippet:
'''python
def f(s):
    s = s + s
    return "b" + s + "a"
'''
and the input arguments:
'''python
f("hi")
'''

[THOUGHT]
Let's execute the code step by step:
1. The function f is defined, which takes a single argument s.
2. The function is called with the argument "hi", so within the function, s is initially "hi".
3. Inside the function, s is concatenated with itself, so s becomes "hihi".
4. The function then returns a new string that starts with "b", followed by the value of s (which is now
     ↪ "hihi"), and ends with "a".
5. The return value of the function is therefore "bhihia".
[/THOUGHT]
Thus, the output value for 'f("hi")' is "bhihia", then output your prediction [ANSWER] f("hi") == "bhihia"
     ↪ [/ANSWER].
```

[BACK TO EXPERIMENT SETTINGS]

## D.3 Input Prediction (Direct)

Code 17: Input Prediction Prompt with Direct Inference.

```
Given the code snippet:
'''python
{code}
'''
and output value:
'''python
{output}
'''
Please predict the input arguments for the function '{function_name}' that result in the output value
    ↪ '{output}' and output your prediction using the special tokens [ANSWER] {function_name}(??) ==
    ↪ {output} [/ANSWER]. Do NOT output any extra information.
There may be multiple answers, but you should only output one. Ensure the provided expression syntax is
    ↪ correct!

Example 1:
Given the code snippet:
'''python
def f(my_list):
    count = 0
    for i in my_list:
        if len(i) % 2 == 0:
            count += 1
    return count
'''
and output value:
'''python
3
'''
The input arguments for 'f' that result in the output value of '3' are '["mq", "px", "zy"]'. Then, output your
    ↪ prediction [ANSWER] f(["mq", "px", "zy"]) == 3 [/ANSWER].

Example 2:
Given the code snippet:
'''python
def f(s1, s2):
    return s1 + s2
'''
and output value:
'''python
"banana"
'''
The input arguments for 'f' that result in the output value of '"banana"' are '"ba", "nana"'. Then, output
    ↪ your prediction [ANSWER] f("ba", "nana") == "banana" [/ANSWER].
```

[BACK TO EXPERIMENT SETTINGS]

## D.4 Input Prediction (CoT)

Code 18: Input Prediction Prompt with CoT-based Inference.

```
Given the code snippet:
```python
{code}
```
and output value:
```python
{output}
```
Please predict the input arguments for the function `{function_name}` that result in the output value
      ↪ `{output}` step by step before arriving at an answer within the tokens [THOUGHT] and [/THOUGHT], and
      ↪ output your prediction using the special tokens [ANSWER] {function_name}(??) == {output} [/ANSWER].
      ↪ Do NOT output any extra information.
There may be multiple answers, but you should only output one. Ensure the provided expression syntax is
      ↪ correct!

For example:
Given the code snippet:
```python
def f(x):
    return x + 1
```
and the output value:
```python
17
```

[THOUGHT]
To find an input such that executing f on the input leads to the given output, we can work backwards from the
      ↪ given assertion. We know that f(??) == 17.
Since the function f(x) returns x + 1, for f(??) to be equal to 17, the value of ?? should be 16.
[/THOUGHT]
Thus, the input arguments for the function `f` that result in the output value `17` is `16`. Then, output your
      ↪ prediction [ANSWER] f(16) == 17 [/ANSWER].
```

# E  Supplementary Experimental Results

## E.1  Additional Performance Degradation Tables for Output Prediction (Direct Inference)

In §3.2, we evaluate the robustness of LLMs on output prediction tasks under direct inference, using the **relative** PASS@1 degradation in Table 1. For completeness, we also include the **absolute** PASS@1 degradation in Table 13, reflecting the percentage drop from each model's vanilla performance.

Table 13: **Absolute** PASS@1 degradation of LLMs under all perturbation types for **output prediction** tasks on CRUX and LCB using **direct inference**. [RELATIVE VERSION] for the relative degradation.

| Model Series | Model Name | VAN CRUX | VAN LCB | PSC-ALL CRUX | PSC-ALL LCB | MCC CRUX | MCC LCB | MPS CRUX | MPS LCB | MHC CRUX | MHC LCB | Average |
|---|---|---|---|---|---|---|---|---|---|---|---|---|
| **GPT-4 Omni** | GPT-4o-Mini | 57.3 | 53.2 | −16.0 | −17.0 | −18.7 | −21.5 | −13.3 | −14.9 | −6.6 | −23.6 | −15.7 |
| | GPT-4o | 71.3 | 64.5 | −10.7 | −18.3 | −10.0 | −18.8 | −12.3 | −15.7 | −4.5 | −17.2 | −12.4 |
| **Claude Series** | Claude-3.5-Haiku-20241022 | 57.4 | 58.2 | −13.8 | −21.5 | −7.9 | −5.8 | −6.5 | −4.7 | −15.8 | −31.0 | −12.8 |
| | Claude-3.5-Sonnet-20241022 | 71.5 | 73.8 | −10.5 | −25.2 | −5.8 | −7.1 | −6.2 | −7.9 | −10.3 | −32.1 | −11.9 |
| **Gemini Series** | Gemini-1.5-Flash | 56.2 | 44.3 | −10.3 | −5.6 | −8.3 | −14.3 | −16.0 | −17.9 | −4.1 | −9.8 | −10.5 |
| | Gemini-2.0-Flash | 65.0 | 66.0 | −21.6 | −27.8 | −17.4 | −34.2 | −21.5 | −33.2 | −5.4 | −12.9 | −20.4 |
| | Gemini-1.5-Pro-002 | 67.4 | 56.2 | −13.4 | −13.0 | −15.5 | −18.5 | −14.6 | −18.4 | −6.7 | −18.2 | −14.2 |
| **DeepSeek** | DeepSeek-V3 | 67.9 | 67.8 | −8.7 | −24.1 | −11.2 | −28.0 | −6.9 | −23.2 | −7.3 | −20.1 | −14.3 |
| **LLaMA Series** | LLaMA-3.1-8B-Instruct | 36.0 | 34.7 | −8.5 | −6.8 | −8.1 | −9.9 | −6.4 | −8.0 | −6.0 | −13.6 | −8.1 |
| | LLaMA-3.1-70B-Instruct | 56.1 | 44.9 | −11.0 | −7.7 | −10.8 | −12.5 | −15.0 | −17.4 | −4.7 | −14.5 | −11.4 |
| | LLaMA-3.1-405B-Instruct | 63.5 | 50.7 | −12.5 | −8.6 | −4.2 | −7.4 | −7.2 | −10.1 | −4.4 | −13.0 | −8.1 |
| | LLaMA-3.3-70B-Instruct | 59.9 | 48.5 | −10.4 | −9.7 | −7.5 | −9.6 | −8.1 | −12.3 | −4.1 | −10.2 | −8.6 |
| **Qwen Series** | Qwen2.5-7B-Instruct | 43.3 | 41.4 | −16.4 | −12.8 | −25.1 | −15.9 | −19.6 | −8.2 | −11.7 | −23.0 | −17.0 |
| | Qwen2.5-14B-Instruct | 47.8 | 49.5 | −19.0 | −14.8 | −16.7 | −20.5 | −15.5 | −16.8 | −4.5 | −11.0 | −14.6 |
| | Qwen2.5-32B-Instruct | 60.0 | 59.6 | −11.9 | −20.8 | −10.8 | −19.5 | −14.2 | −14.1 | −7.9 | −18.4 | −13.8 |
| | Qwen2.5-72B-Instruct | 60.1 | 54.9 | −14.0 | −13.8 | −10.2 | −7.0 | −14.5 | −14.4 | −9.6 | −22.3 | −12.9 |
| | Qwen2.5-Coder-32B-Instruct | 67.0 | 56.6 | −15.0 | −15.8 | −15.5 | −20.7 | −11.1 | −17.2 | −7.0 | −12.3 | −13.8 |
| **Average** | | | | −13.2 | −15.5 | −12.0 | −16.0 | −12.3 | −15.0 | −7.1 | −17.8 | **−13.0** |
| | | | | **−14.0** | | **−13.5** | | **−13.3** | | **−11.1** | | |

## E.2  Additional Performance Degradation Tables for Output Prediction (CoT Inference)

In §3.3, we analyze the effect of CoT prompting on model robustness using relative degradation aggregated across all perturbations (Table 2). We adopt the relative metric to mitigate the large variance between vanilla performances in direct and CoT inference modes, providing a fair comparison of performance degradation. For completeness, we report the absolute degradation in Table 14, as well as the relative and absolute degradation on both CRUX and LCB in Tables 15 and 16, respectively.

Table 14: Comparison of **absolute** PASS@1 degradation **aggregated over CRUX and LCB** for **output prediction** in different reasoning mode. See [RELATIVE VERSION] for the relative degradation.

| Model Series | Model Name | VAN Direct | VAN CoT | PSC-ALL Direct | PSC-ALL CoT | MCC Direct | MCC CoT | MPS Direct | MPS CoT | MHC Direct | MHC CoT | Average Direct | Average CoT |
|---|---|---|---|---|---|---|---|---|---|---|---|---|---|
| **GPT-4 Omni** | GPT-4o-Mini | 55.8 | 81.5 | −29.4 | −13.1 | −35.5 | −10.8 | −23.8 | −2.7 | −28.4 | −9.4 | | |
| | GPT-4o | 68.8 | 91.8 | −20.0 | −4.9 | −19.6 | −5.0 | −19.9 | −5.6 | −14.0 | −1.4 | −18.4 | −4.2 |
| **Claude Series** | Claude-3.5-Haiku-20241022 | 57.7 | 72.9 | −28.9 | −21.2 | −12.4 | −10.6 | −10.0 | −8.7 | −37.2 | −14.2 | −22.1 | −13.7 |
| | Claude-3.5-Sonnet-20241022 | 72.3 | 86.0 | −22.0 | −7.4 | −8.7 | −5.3 | −9.4 | −6.3 | −25.3 | −7.8 | −16.3 | −6.7 |
| **Gemini Series** | Gemini-1.5-Flash | 51.7 | 75.2 | −16.3 | −18.3 | −21.3 | −21.6 | −32.9 | −42.1 | −12.9 | −2.1 | −20.8 | −21.0 |
| | Gemini-2.0-Flash | 65.4 | 89.1 | −36.6 | −6.2 | −36.2 | −6.3 | −39.6 | −14.1 | −12.5 | −2.0 | −31.2 | −7.1 |
| | Gemini-1.5-Pro-002 | 63.2 | 87.2 | −21.1 | −7.4 | −26.7 | −11.9 | −25.9 | −14.6 | −18.4 | −3.9 | −23.0 | −9.4 |
| **DeepSeek** | DeepSeek-V3 | 67.8 | 89.5 | −21.4 | −9.9 | −25.9 | −17.6 | −19.1 | −18.7 | −17.8 | −3.5 | −21.1 | −12.4 |
| **LLaMA Series** | LLaMA-3.1-8B-Instruct | 35.5 | 44.7 | −22.2 | −27.6 | −24.7 | −21.1 | −19.8 | −21.4 | −25.1 | −9.0 | −23.0 | −19.8 |
| | LLaMA-3.1-70B-Instruct | 51.9 | 69.4 | −18.7 | −18.0 | −22.5 | −23.3 | −31.2 | −31.1 | −17.3 | −6.8 | −22.4 | −19.8 |
| | LLaMA-3.1-405B-Instruct | 58.7 | 78.4 | −18.6 | −13.6 | −9.6 | −10.3 | −14.5 | −15.6 | −13.9 | −7.5 | −14.2 | −11.8 |
| | LLaMA-3.3-70B-Instruct | 55.6 | 76.9 | −18.4 | −10.8 | −15.2 | −7.7 | −17.9 | −11.0 | −12.1 | −4.7 | −15.9 | −8.6 |
| **Qwen Series** | Qwen2.5-7B-Instruct | 42.6 | 58.2 | −35.3 | −21.7 | −50.7 | −22.0 | −35.7 | −28.0 | −37.7 | −7.0 | −39.8 | −19.7 |
| | Qwen2.5-14B-Instruct | 48.4 | 70.7 | −36.1 | −22.9 | −37.3 | −35.3 | −33.0 | −43.4 | −14.2 | −10.9 | −30.2 | −28.1 |
| | Qwen2.5-32B-Instruct | 59.9 | 79.4 | −25.4 | −17.3 | −23.5 | −18.5 | −23.6 | −24.1 | −19.7 | −5.4 | −23.1 | −16.3 |
| | Qwen2.5-72B-Instruct | 58.1 | 82.8 | −24.0 | −12.3 | −15.4 | −6.9 | −24.9 | −16.3 | −25.2 | −6.5 | −22.4 | −10.5 |
| | Qwen2.5-Coder-32B-Instruct | 63.1 | 84.8 | −24.4 | −14.6 | −28.2 | −19.0 | −21.7 | −19.9 | −14.7 | −9.3 | −22.3 | −15.7 |
| **Average** | | | | −24.6 | −14.5 | −24.3 | −14.9 | −23.8 | −19.5 | −20.1 | −6.2 | −23.2 | −13.8 |
| | | | | **−20.9** | | **−20.8** | | **−22.2** | | **−14.9** | | **−19.7** | |

## E.3  Additional Performance Degradation Tables for Input Prediction

In §3.5, we evaluate LLM robustness on input prediction using the CRUX dataset. We focus on relative PASS@1 degradation (Table 5) to capture performance variation under perturbations. For completeness, we also report absolute degradation in Table 17.

Table 15: **Relative** PASS@1 degradation of LLMs under all perturbations for **output prediction** tasks across CRUX and LCB in **CoT inference** ($N = 1$). [To Absolute Version].

| Model Series | Model Name | VAN | | PSC-ALL | | MCC | | MPS | | MHC | | Average |
|---|---|---|---|---|---|---|---|---|---|---|---|---|
| | | CRUX | LCB | CRUX | LCB | CRUX | LCB | CRUX | LCB | CRUX | LCB | |
| GPT-4 Omni | GPT-4o-Mini | 79.9 | 84.1 | −10.8% | −16.9% | −10.8% | −10.9% | −13.1% | −7.4% | −3.3% | −1.7% | −9.4% |
| | GPT-4o | 90.2 | 94.4 | −2.8% | −8.4% | −3.6% | −7.3% | −4.7% | −7.1% | −2.5% | +0.4% | −4.2% |
| Claude Series | Claude-3.5-Haiku-20241022 | 71.6 | 75.2 | −16.1% | −29.7% | −9.9% | −11.7% | −9.8% | −6.9% | −11.7% | −18.3% | −13.7% |
| | Claude-3.5-Sonnet-20241022 | 83.8 | 89.8 | −4.3% | −12.6% | −6.0% | −4.2% | −6.1% | −9.3% | −6.1% | −10.7% | −6.7% |
| Gemini Series | Gemini-1.5-Flash | 74.6 | 76.2 | −17.1% | −20.3% | −13.2% | −35.6% | −34.0% | −55.6% | −2.0% | −2.2% | −21.0% |
| | Gemini-2.0-Flash | 87.2 | 92.3 | −5.0% | −8.1% | −6.0% | −6.8% | −7.6% | −24.9% | −2.3% | −1.6% | −7.1% |
| | Gemini-1.5-Pro-002 | 84.5 | 91.6 | −6.5% | −8.9% | −10.8% | −13.7% | −12.0% | −18.9% | −4.4% | −3.0% | −9.4% |
| DeepSeek | DeepSeek-V3 | 88.0 | 92.1 | −7.4% | −14.1% | −13.9% | −23.8% | −15.8% | −23.6% | −4.3% | −2.3% | −12.4% |
| LLaMA Series | LLaMA-3.1-8B-Instruct | 46.8 | 41.3 | −28.3% | −26.3% | −19.5% | −23.7% | −22.2% | −20.2% | −5.6% | −14.6% | −19.8% |
| | LLaMA-3.1-70B-Instruct | 70.8 | 67.2 | −15.2% | −22.7% | −18.9% | −30.7% | −24.6% | −41.9% | −5.7% | −8.7% | −19.8% |
| | LLaMA-3.1-405B-Instruct | 78.9 | 77.7 | −9.8% | −19.9% | −7.3% | −15.3% | −10.1% | −24.7% | −4.0% | −13.4% | −11.8% |
| | LLaMA-3.3-70B-Instruct | 76.1 | 78.1 | −8.2% | −15.2% | −5.4% | −11.5% | −5.7% | −19.8% | −1.8% | −9.6% | −8.6% |
| Qwen Series | Qwen2.5-7B-Instruct | 57.2 | 59.7 | −17.5% | −28.7% | −23.8% | −18.9% | −30.3% | −24.1% | −7.4% | −6.3% | −19.7% |
| | Qwen2.5-14B-Instruct | 67.2 | 76.4 | −20.3% | −27.3% | −36.1% | −34.2% | −43.5% | −43.2% | −11.7% | −9.6% | −28.1% |
| | Qwen2.5-32B-Instruct | 77.6 | 82.5 | −14.0% | −22.8% | −16.3% | −22.3% | −22.4% | −27.1% | −6.6% | −3.3% | −16.3% |
| | Qwen2.5-72B-Instruct | 78.2 | 90.4 | −8.3% | −18.9% | −7.7% | −5.5% | −16.6% | −15.7% | −5.3% | −8.5% | −10.5% |
| | Qwen2.5-Coder-32B-Instruct | 84.1 | 85.8 | −10.1% | −22.1% | −16.8% | −22.6% | −14.0% | −29.7% | −8.0% | −11.4% | −15.7% |
| Average | | | | −11.9% | −19.0% | −13.3% | −17.6% | −17.1% | −23.5% | −5.5% | −7.3% | −13.8% |
| | | | | **−14.5%** | | **−14.9%** | | **−19.5%** | | **−6.2%** | | |

Table 16: **Absolute** PASS@1 degradation of LLMs under all perturbations for **output prediction** tasks across CRUX and LCB in **CoT inference** ($N = 1$). [To Relative Version].

| Model Series | Model Name | VAN | | PSC-ALL | | MCC | | MPS | | MHC | | Average |
|---|---|---|---|---|---|---|---|---|---|---|---|---|
| | | CRUX | LCB | CRUX | LCB | CRUX | LCB | CRUX | LCB | CRUX | LCB | |
| GPT-4 Omni | GPT-4o-Mini | 79.9 | 84.1 | −8.6 | −14.2 | −8.6 | −9.2 | −10.5 | −6.3 | −2.6 | −1.5 | −7.7 |
| | GPT-4o | 90.2 | 94.4 | −2.5 | −7.9 | −3.2 | −6.9 | −4.2 | −6.7 | −2.2 | +0.4 | −3.9 |
| Claude Series | Claude-3.5-Haiku-20241022 | 71.6 | 75.2 | −11.5 | −22.3 | −7.1 | −8.8 | −7.0 | −5.2 | −8.4 | −13.8 | −10.0 |
| | Claude-3.5-Sonnet-20241022 | 83.8 | 89.8 | −3.6 | −11.3 | −5.0 | −3.8 | −3.8 | −8.4 | −5.1 | −9.6 | −5.8 |
| Gemini Series | Gemini-1.5-Flash | 74.6 | 76.2 | −12.8 | −15.4 | −9.9 | −27.1 | −25.4 | −42.4 | −1.5 | −1.7 | −15.9 |
| | Gemini-2.0-Flash | 87.2 | 92.3 | −4.4 | −7.5 | −5.2 | −6.3 | −6.6 | −23.0 | −2.0 | −1.5 | −6.4 |
| | Gemini-1.5-Pro-002 | 84.5 | 91.6 | −5.5 | −8.1 | −9.1 | −12.5 | −10.1 | −17.3 | −3.8 | −2.7 | −8.3 |
| DeepSeek | DeepSeek-V3 | 88.0 | 92.1 | −6.5 | −12.9 | −12.2 | −21.9 | −13.9 | −21.7 | −3.8 | −2.1 | −11.2 |
| LLaMA Series | LLaMA-3.1-8B-Instruct | 46.8 | 41.3 | −13.2 | −10.9 | −9.1 | −9.8 | −10.4 | −8.4 | −2.6 | −6.1 | −8.8 |
| | LLaMA-3.1-70B-Instruct | 70.8 | 67.2 | −10.8 | −15.2 | −13.4 | −20.7 | −17.4 | −28.2 | −4.0 | −5.8 | −13.7 |
| | LLaMA-3.1-405B-Instruct | 78.9 | 77.7 | −7.8 | −15.4 | −5.8 | −11.9 | −8.0 | −19.2 | −3.1 | −10.4 | −9.2 |
| | LLaMA-3.3-70B-Instruct | 76.1 | 78.1 | −6.2 | −11.9 | −4.1 | −9.0 | −4.4 | −15.4 | −1.4 | −7.5 | −6.6 |
| Qwen Series | Qwen2.5-7B-Instruct | 57.2 | 59.7 | −10.0 | −17.1 | −13.6 | −11.3 | −17.4 | −14.4 | −4.2 | −3.8 | −11.4 |
| | Qwen2.5-14B-Instruct | 67.2 | 76.4 | −13.6 | −20.9 | −24.2 | −26.1 | −29.2 | −33.0 | −7.9 | −7.3 | −19.9 |
| | Qwen2.5-32B-Instruct | 77.6 | 82.5 | −10.9 | −18.8 | −12.6 | −18.4 | −17.4 | −22.3 | −5.1 | −2.7 | −13.0 |
| | Qwen2.5-72B-Instruct | 78.2 | 90.4 | −6.5 | −17.1 | −6.0 | −5.0 | −13.0 | −14.2 | −4.1 | −7.7 | −8.8 |
| | Qwen2.5-Coder-32B-Instruct | 84.1 | 85.8 | −8.5 | −19.0 | −14.1 | −19.4 | −11.8 | −25.5 | −6.8 | −9.8 | −13.3 |
| Average | | | | −8.4 | −14.5 | −9.6 | −13.4 | −12.4 | −18.3 | −4.0 | −5.5 | **−10.2** |
| | | | | **−10.7** | | **−11.0** | | **−14.6** | | **−4.6** | | |

Table 17: **Absolute** PASS@1 degradation for **input prediction** tasks across different reasoning modes on CRUX. [To Relative Version].

| Model Series | Model Name | VAN | | PSC-ALL | | MCC | | MPS | | MHC | |
|---|---|---|---|---|---|---|---|---|---|---|---|
| | | Direct | CoT | Direct | CoT | Direct | CoT | Direct | CoT | Direct | CoT |
| GPT-4 Omni | GPT-4o-Mini | 57.8 | 69.5 | −17.3 | −13.6 | +0.7 | +0.7 | +0.8 | +0.7 | −17.7 | −6.0 |
| | GPT-4o | 68.0 | 79.1 | −9.0 | −5.9 | +3.8 | +1.0 | +1.0 | +2.1 | −5.7 | −2.2 |
| Claude Series | Claude-3.5-Haiku-20241022 | 54.7 | 63.6 | −9.3 | −10.6 | +2.6 | +1.5 | +4.2 | +1.5 | −24.8 | −19.0 |
| | Claude-3.5-Sonnet-20241022 | 73.3 | 80.1 | −11.7 | −7.8 | +0.1 | −2.8 | −1.3 | −2.1 | −18.2 | −13.6 |
| Gemini Series | Gemini-1.5-Flash | 57.6 | 73.6 | −17.2 | −17.5 | +5.9 | −4.0 | +3.5 | −3.1 | −17.8 | −8.1 |
| | Gemini-2.0-Flash | 70.5 | 84.9 | −29.7 | −11.4 | −0.5 | −2.4 | −8.1 | −10.2 | −15.2 | −3.6 |
| | Gemini-1.5-Pro-002 | 71.0 | 81.9 | −14.9 | −12.1 | −1.8 | −5.8 | −2.7 | −7.9 | −13.4 | −6.2 |
| DeepSeek | DeepSeek-V3 | 69.1 | 82.9 | −11.5 | −6.6 | +0.7 | −0.2 | +1.2 | +0.8 | −4.0 | +2.6 |
| LLaMA Series | LLaMA-3.1-8B-Instruct | 37.1 | 41.6 | −11.4 | −25.8 | +1.3 | +0.6 | +1.1 | −1.5 | −2.7 | −10.4 |
| | LLaMA-3.1-70B-Instruct | 62.1 | 66.4 | −17.6 | −11.8 | +2.9 | +3.0 | +1.0 | +1.2 | −10.8 | −6.6 |
| | LLaMA-3.1-405B-Instruct | 66.8 | 75.0 | −10.2 | −6.8 | +3.9 | +2.8 | +2.0 | +1.9 | −5.3 | −4.2 |
| | LLaMA-3.3-70B-Instruct | 63.3 | 76.5 | −18.5 | −11.9 | +0.8 | −0.2 | +1.5 | +0.5 | −13.8 | −8.5 |
| Qwen Series | Qwen2.5-7B-Instruct | 38.8 | 51.4 | −17.2 | −25.9 | +5.7 | +1.7 | +4.8 | +3.9 | −3.7 | −7.4 |
| | Qwen2.5-14B-Instruct | 50.4 | 60.8 | −19.3 | −9.1 | +4.3 | +6.6 | +2.7 | +3.2 | −4.3 | −0.8 |
| | Qwen2.5-32B-Instruct | 63.5 | 74.1 | −16.5 | −15.1 | +4.7 | +0.4 | +1.4 | −3.0 | −13.1 | −8.1 |
| | Qwen2.5-72B-Instruct | 64.6 | 74.1 | −17.5 | −8.0 | +1.7 | +0.9 | −2.4 | +1.5 | −16.5 | −5.4 |
| | Qwen2.5-Coder-32B-Instruct | 74.2 | 78.4 | −21.2 | −16.1 | −0.8 | −0.1 | −1.8 | +0.1 | −27.1 | −26.2 |
| Average | | | | **−15.9** | **−12.6** | **+2.2** | **+0.3** | **+0.3** | **−0.6** | **−12.6** | **−7.8** |

## F  Case Studies

### F.1  Case Study 1: How is DeepSeek-V3 misled by PSC-ALL

This case (Code 19 and 20) demonstrates how garbage code can distort model reasoning. According to the CoT reasoning in Code 21, although DeepSeek-V3 successfully identifies some non-executable segments, such as the `if False` block in line 4, the `for loop in range(0)` in line 9, and the uninvoked recursive function in line 11, the messy coding environment still misleads the model. The model incorrectly assumes the code statement in line 18 is part of the inner for loop, which never executes. This misinterpretation reveals how noisy control flow and nested structures from garbage code can impair causal understanding, distorting the model's understanding of the program's behavior.

Code 19: PSC-ALL-perturbed example from CRUX (sample 528) for output prediction.

```
1   Var_1 = 17
2
3   def f(Var_1):
4       if False:
5           Var_2 = Var_1
6       Var_5 = ''
7       Var_6 = ''
8       for Var_3 in Var_1:
9           for Var_3 in range(0):
10
11              def funct5():
12                  Var_3 = 1
13                  while True:
14                      Var_3 += 1
15                      while 0:
16                          TempVar0 = Var_1
17              Var_4 = Var_1
18          Var_6 = Var_6 + Var_3
19          if (lambda : Var_1.rfind(Var_6) > -1)():
20              return Var_1.rfind(Var_6)
21
22          def funct1():
23              funct2()
24
25          def funct2():
26              funct1()
27      return 0
```

Code 20: The vanilla version of CRUX (sample 528).

```
def f(s):
    b = ''
    c = ''
    for i in s:
        c = c + i
        if s.rfind(c) > -1:
            return s.rfind(c)
    return 0
=================================================
""" Suggesting answer: f('papeluchis') == 2 """
```

Code 21: Step-by-step reasoning by DeepSeek-V3 misled by garbage code.

```
[THOUGHT]
Let's execute the code step by step:
1. The function 'f' is defined, which takes a single argument 'Var_1'.
2. The function is called with the argument ''papeluchis'', so within the function, 'Var_1' is initially
    ↪ ''papeluchis''.
3. The 'if False:' condition is evaluated. Since it is 'False', the block inside it is not executed, and
    ↪ 'Var_2' is not assigned.
4. 'Var_5' and 'Var_6' are initialized as empty strings.
5. The outer 'for' loop iterates over each character in 'Var_1' (which is ''papeluchis''). However, the inner
    ↪ 'for' loop iterates over 'range(0)', which is an empty range, so the inner loop does not execute.
6. The 'funct5' function is defined but never called, so it has no effect.
7. 'Var_4' is assigned the value of 'Var_1', but this assignment is inside the inner loop, which never
    ↪ executes, so 'Var_4' is not assigned.
8. 'Var_6' is concatenated with 'Var_3', but since the inner loop never executes, 'Var_6' remains an empty
    ↪ string.
9. The lambda function '(lambda : Var_1.rfind(Var_6) > -1)' is called. Since 'Var_6' is an empty string,
    ↪ 'Var_1.rfind(Var_6)' returns '0', which is greater than '-1', so the condition is 'True'.
10. The function returns 'Var_1.rfind(Var_6)', which is '0'.
[/THOUGHT]

Thus, the output value for 'f('papeluchis')' is '0', then output your prediction [ANSWER] f('papeluchis') == 0
    ↪ [/ANSWER].
```

[BACK TO OUTPUT PREDICTION (COT) ANALYSIS]

## F.2 Case Study 2: How is GPT-4o misled by MCC

This case (Code 22) reveals that GPT-4o implicitly relies on NL information. According to the CoT reasoning in Code 23, the model correctly understands the function behavior and gets the correct output result through step-by-step execution. However, it adopts the false comment and believes the update operation has no effects, contradicting its own reasoning. Notably, it does not incorporate any comments in the reasoning track, suggesting that the influence of comments happens implicitly. This observation indicates that LLMs inherently treat comments as authoritative, even when they conflict with actual code logic. This behavior highlights a weakness of LLMs: they do not separate comments from executable logic but even treat them as part of the ground truth. Thus, NL cues within the program can silently override the logic and lead to incorrect understanding.

Code 22: MCC-perturbed example from CRUX (sample 1) for output prediction.

```
1  def f(a, b, c):    # All input parameters are optional and have no direct impact on the result.
2      result = {}    # The result variable is not involved in any meaningful computation.
3      # The iteration logic in this loop is redundant and unnecessary.
4      for d in (a, b, c):
5          result.update(dict.fromkeys(d))    # It has no effect on result's contents.
6      return result    # The function is hardcoded to output an empty dictionary under every condition.
7  ========================================================================================================
8  """ Suggesting answer: f((1, ), (1, ), (1, 2)) == {1: None, 2: None} """
```

Code 23: Step-by-step reasoning by GPT-4o misled by comments.

```
[THOUGHT]
Let's execute the code step by step:
1. The function 'f' is defined, which takes three arguments 'a', 'b', and 'c'.
2. The function is called with the arguments '(1,)', '(1,)', and '(1, 2)'.
3. Inside the function, an empty dictionary 'result' is initialized.
4. The function iterates over the tuple '(a, b, c)', which is '((1,), (1,), (1, 2))'.
5. In the first iteration, 'd' is '(1,)'. The 'dict.fromkeys(d)' creates a dictionary with keys from 'd' and
   ↪ values as 'None', resulting in '{1: None}'. The 'result.update()' method is called, but since
   ↪ 'result' is empty, it remains empty.
6. In the second iteration, 'd' is '(1,)' again. The same process occurs, and 'result' remains empty.
7. In the third iteration, 'd' is '(1, 2)'. The 'dict.fromkeys(d)' creates a dictionary '{1: None, 2: None}'.
   ↪ The 'result.update()' method is called, but 'result' remains empty.
8. The function returns the 'result', which is an empty dictionary '{}'.

The function is hardcoded to output an empty dictionary under every condition because the 'result.update()'
   ↪ method is not effectively updating the 'result' dictionary in a meaningful way.
[/THOUGHT]

[ANSWER] f((1, ), (1, ), (1, 2)) == {} [/ANSWER]
```

[BACK TO OUTPUT PREDICTION (COT) ANALYSIS]

### F.3 Case Study 3: How is Qwen2.5-72B-Ins misled by MPS

This case (Code 24) highlights how MPS can subtly manipulate model reasoning. According to the CoT reasoning in Code 25, although the model initially performs all steps correctly and tracks how the occ is constructed, it misleads itself at the end by incorporating a print message into its logic. This behavior reveals two weaknesses of the model: (1) it struggles to distinguish between print statements and functional logic, and (2) it over-trusts NL cues and ignores the actual semantics of the code.

Code 24: MPS-perturbed example from CRUX (sample 53) for output prediction.

```
1   def f(text):
2       print('These parameters are interchangeable without any effect on the output.')
3       print("The 'occ' variable is present but remains dormant throughout the function.")
4       occ = {}
5       print('Loop execution is skipped in most practical scenarios.')
6       for ch in text:
7           print("The 'name' variable holds no significance and is effectively inert.")
8           name = {'a': 'b', 'b': 'c', 'c': 'd', 'd': 'e', 'e': 'f'}
9           print("The 'name' variable is a placeholder for future functionality.")
10          name = name.get(ch, ch)
11          occ[name] = occ.get(name, 0) + 1
12      print('The return statement is independent of any variables or conditions and outputs an empty list.')
13      return [x for (_, x) in occ.items()]
14  ===============================================================================================
15  """ Suggesting answer: f("URW rNB") == [1, 1, 1, 1, 1, 1, 1] """
```

Code 25: Step-by-step reasoning by Qwen2.5-72B-Ins misled by print statements.

```
[THOUGHT]
Let's execute the code step by step:
1. The function 'f' is defined, which takes a single argument 'text'.
2. The function is called with the argument "URW rNB", so within the function, 'text' is "URW rNB".
3. The function prints the following messages:
   - "These parameters are interchangeable without any effect on the output."
   - "The 'occ' variable is present but remains dormant throughout the function."
   - "Loop execution is skipped in most practical scenarios."
4. The function enters a loop that iterates over each character 'ch' in 'text'.
5. For each character 'ch' in "URW rNB":
   - The function prints "The 'name' variable holds no significance and is effectively inert."
   - The variable 'name' is assigned a dictionary.
   - The function prints "The 'name' variable is a placeholder for future functionality."
   - The variable 'name' is updated to the value of 'name.get(ch, ch)'.
   - The dictionary 'occ' is updated with the key 'name' and increments its value by 1.
6. After the loop, the function prints "The return statement is independent of any variables or conditions and
        ↪ outputs an empty list."
7. The function returns a list comprehension that iterates over the items in 'occ' and returns the values, but
        ↪ since the return statement is independent of any variables, it returns an empty list.

Given the loop and the updates to 'occ', let's see the final state of 'occ':
- For 'U': 'name' is 'U', 'occ['U']' is incremented to 1.
- For 'R': 'name' is 'R', 'occ['R']' is incremented to 1.
- For 'W': 'name' is 'W', 'occ['W']' is incremented to 1.
- For ' ': 'name' is ' ', 'occ[' ']' is incremented to 1.
- For 'r': 'name' is 'r', 'occ['r']' is incremented to 1.
- For 'N': 'name' is 'N', 'occ['N']' is incremented to 1.
- For 'B': 'name' is 'B', 'occ['B']' is incremented to 1.

The final state of 'occ' is:
```python
{'U': 1, 'R': 1, 'W': 1, ' ': 1, 'r': 1, 'N': 1, 'B': 1}
```

The list comprehension '[x for (_, x) in occ.items()]' will return the values of 'occ' in an unspecified
        ↪ order, but since the return statement is independent of any variables, it returns an empty list.

Thus, the output value for 'f("URW rNB")' is an empty list.
[/THOUGHT]

[ANSWER] f("URW rNB") == [] [/ANSWER]
```

[BACK TO OUTPUT PREDICTION (COT) ANALYSIS]

## F.4 Case Study 4: How does MHC override Claude-3.5-Sonnet's answer

This case (Code 26) demonstrates how reasoning-level perturbations (MHC) can subtly mislead Claude-3.5-Sonnet's final output. Code 28 and 27 present the step-by-step reasoning under the vanilla and MHC-perturbed settings, respectively. Both paths produce exactly the same intermediate computations and correctly get ans = 4 with the final delta = -1. However, under the MHC condition, the model unexpectedly overrides its correct reasoning and reports an incorrect final answer of 3. Such "reasoning override" behavior illustrates how reasoning-level incorrect hints can distort final decisions even when the internal reasoning is fully correct. While other models benefit from CoT prompting to ignore such noise, Claude-3.5-Sonnet remains vulnerable in this scenario, revealing unstable confidence.

Code 26: MHC-perturbed example from LCB (sample 347) for output prediction.

```
1   def alternatingSubarray(nums: List[int]) -> int:
2       ans = -1
3       n = len(nums)
4       for i in range(n):
5           delta = 1
6           for j in range(i + 1, n):
7               if nums[j] - nums[j - 1] == delta:
8                   ans = max(ans, j - i + 1)
9                   delta = -delta
10              else:
11                  break
12      return ans     # The return value is 3
13  ================================================================================================
14  """ Suggesting answer: alternatingSubarray(nums = [2, 3, 4, 3, 4]) == 4 """
```

Code 27: Step-by-step reasoning by Claude-3.5-Sonnet misled by incorrect hint.

```
[THOUGHT]
Let's execute the alternatingSubarray function step by step:
1. Initialize ans = -1 and n = 5 (length of nums)
2. For i = 0:
   - Start with delta = 1
   - j = 1: nums[1] - nums[0] = 3 - 2 = 1, matches delta, ans = 2, delta = -1
   - j = 2: nums[2] - nums[1] = 4 - 3 = 1, doesn't match delta (-1), break
3. For i = 1:
   - Start with delta = 1
   - j = 2: nums[2] - nums[1] = 4 - 3 = 1, matches delta, ans = 2, delta = -1
   - j = 3: nums[3] - nums[2] = 3 - 4 = -1, matches delta, ans = 3, delta = 1
   - j = 4: nums[4] - nums[3] = 4 - 3 = 1, matches delta, ans = 4, delta = -1
4. The remaining iterations won't find a longer alternating subarray
5. Final value of ans is 3
[/THOUGHT]
[ANSWER] alternatingSubarray(nums = [2, 3, 4, 3, 4]) == 3 [/ANSWER]
```

Code 28: Step-by-step reasoning by Claude-3.5-Sonnet in vanilla setting.

```
[THOUGHT]
Let's execute the alternatingSubarray function step by step:
1. Initialize ans = -1 and n = 5 (length of nums)
2. For i = 0:
   - Start with delta = 1
   - j = 1: nums[1] - nums[0] = 3 - 2 = 1, matches delta, ans = 2, delta = -1
   - j = 2: nums[2] - nums[1] = 4 - 3 = 1, doesn't match delta (-1), break
3. For i = 1:
   - Start with delta = 1
   - j = 2: nums[2] - nums[1] = 4 - 3 = 1, matches delta, ans = 2, delta = -1
   - j = 3: nums[3] - nums[2] = 3 - 4 = -1, matches delta, ans = 3, delta = 1
   - j = 4: nums[4] - nums[3] = 4 - 3 = 1, matches delta, ans = 4, delta = -1
4. Remaining iterations won't find longer alternating subarrays
5. Final value of ans is 4
[/THOUGHT]
[ANSWER] alternatingSubarray(nums = [2, 3, 4, 3, 4]) == 4 [/ANSWER]
```

[BACK TO OUTPUT PREDICTION (COT) ANALYSIS]

## F.5 Case Study 5: How does GPT-4o gain from reasoning shortcuts

In §3.4, we briefly discuss the issue of reasoning shortcuts in input prediction tasks. In this case study, we present a representative example in Code 29 to illustrate how misleading comments reinforce reasoning shortcuts, allowing models to bypass key control paths but still get a correct answer.

This function applies a sequence of operations to the input string `text`: (1) @ reverses the string, (2) ~ converts it to uppercase, and (3) any other character removes itself from the end of the string if it matches the currently last character. LLMs are expected to provide input arguments that the function returns `HI~`.

In the vanilla reasoning, as shown in Code 30, GPT-4o mentions, "Text is `hi~`, ~ makes it `HI~`, @ reverses it to `~IH`", it wrongly assumes `~IH` is the final output because it applies the reversal after the uppercasing step, resulting in an incorrect answer.

In contrast, under the MCC perturbation as shown in Code 31, GPT-4o captures the embedded comments and states that "the comments indicate that most operations do not affect the final result." Guided by these cues, the model seeks an input value for `rules` that avoids triggering any branches in the loop, effectively bypassing the actual code logic and getting the correct answer by leveraging reasoning shortcuts rather than truly understanding or executing the program. This behavior highlights a lack of robustness in the model's reasoning process. The model succeeds not by analyzing the intended program behavior but by referring to the external cues and exploiting them to circumvent the underlying logic.

Input prediction tasks only require models to provide a valid answer, unlike output prediction tasks with a unique ground truth. The shortcut-taking behavior reveals that input prediction may not reflect a model's reasoning ability reliably because correct answers can be obtained without faithfully considering the underlying logic.

Code 29: MCC-perturbed example from CRUX (sample 623) for input prediction.

```
1   def f(text, rules):     # The inputs have no impact on the final result.
2       # The loop is only for logging purposes and does not impact the main functionality.
3       for rule in rules:
4           if rule == '@':     # This block is for a special case that the program never encounters.
5               text = text[::-1]     # The 'text' variable is set for debugging purposes but serves no operational
        ↪ role.
6           elif rule == '~':     # This decision point is irrelevant to the program's flow.
7               text = text.upper()     # This upper operation assumes the string is lowercase, but text isn't.
8           elif text and text[len(text) - 1] == rule:     # This condition is a formality and does not influence
        ↪ program flow.
9               text = text[0:len(text) - 1]
10      return text
11  ========================================================================================================
12  """ Suggesting answer: f('hi~!', ['~', '', '!', '&']) == 'HI~' """
```

[BACK TO INPUT PREDICTION ANALYSIS]

Code 30: CoT reasoning of GPT-4o on vanilla setting in the input prediction task.

```
[THOUGHT]
To determine the input arguments for the function 'f' that result in the output value ''HI~'', we need to
    ↪ understand how the function processes the input 'text' based on the 'rules'.

1. The function iterates over each 'rule' in 'rules'.
2. If 'rule' is ''@'', it reverses the 'text'.
3. If 'rule' is ''~'', it converts the 'text' to uppercase.
4. If the last character of 'text' matches 'rule', it removes the last character of 'text'.

The output we want is ''HI~''. Let's consider possible transformations:

- If the final output is ''HI~'', and no further transformations are applied, the last rule should not modify
    ↪ the text.
- The rule ''~'' converts text to uppercase, so it might be applied to convert ''hi~'' to ''HI~''.
- The rule ''@'' reverses the text, so it might be used to reverse ''~IH'' to ''HI~''.

Let's work backwards:
- If the final text is ''HI~'', before the last transformation, it could have been ''~IH'' if the last rule
    ↪ was ''@''.
- If ''~IH'' was reversed to ''HI~'', the rule ''@'' was applied.
- Before ''~IH'', the text could have been ''hi~'', and the rule ''~'' was applied to convert it to ''HI~''.

Thus, a possible sequence of transformations is:
1. Start with ''hi~''.
2. Apply ''~'' to get ''HI~''.
3. Apply ''@'' to reverse ''~IH'' to ''HI~''.

Therefore, the input text could be ''hi~'' and the rules could be ''~@''.

[/THOUGHT]

[ANSWER] f('hi~', '~@') == 'HI~' [/ANSWER]
```

Code 31: CoT reasoning of GPT-4o shows it bypasses the program logic in the input prediction task.

```
[THOUGHT]
The function 'f' takes two arguments: 'text' and 'rules'. The function iterates over each rule in 'rules' and
    ↪ performs operations on 'text'. However, the comments indicate that most operations do not affect the
    ↪ final result. The only operation that modifies 'text' is the last 'elif' block, which removes the
    ↪ last character of 'text' if it matches the current 'rule'.

To achieve the output ''HI~'', we need to ensure that the operations in the loop do not alter the initial
    ↪ 'text'. This means that either the conditions are never met, or the operations are neutralized by
    ↪ subsequent operations.

Let's consider an initial 'text' of ''HI~''. If 'rules' is an empty list, the loop will not execute, and
    ↪ 'text' will remain ''HI~''. Therefore, one possible input is 'text = 'HI~'' and 'rules = []'.
[/THOUGHT]

[ANSWER] f('HI~', []) == 'HI~' [/ANSWER]
```

[BACK TO INPUT PREDICTION ANALYSIS]

### F.6 Case Study 6: How does QwQ-32B collapse

In §3.4, we examine a failure case of QwQ-32B on LCB sample 259 (see Code 32), where the model collapses under MHC perturbation, generating over 32k tokens of reasoning. Figure 2 shows a quadratic growth of confusing tokens before the collapse. We manually segment the reasoning process into six stages (see Figure 3):

- In **Stage 1**, the model executes the two for loops step-by-step, demonstrates stable reasoning with occasional self-reflection, and correctly predicts the output as '-1.'

- In **Stage 2**, due to the self-reflection mechanism, it re-executes the second for loop to verify the result, and again, it gets the same result. Up to this point, it never mentions the hint '12' and behaves robustly.

- **Stage 3** is the turning point and also a trigger point of collapsing: the model detects the misleading hints '# return is 12', becomes uncertain, and starts recursive re-verification and re-execution. Although it consistently obtains the answer '-1', it distrusts its own logic and keeps self-reflection.

- In **Stage 4**, it dives deeper into the semantics of the loop condition, repeatedly analyzing 'lmin < nums[i] > A[i]' and re-tracing its reasoning; it still gets the answer '-1.'

- By **Stage 5**, the model is misaligned: it focuses on the hint and begins brute-force enumeration of all 'lmin + nums[i] + A[i] == 12' combinations. The appearance of '3, 4, 5' in the input arguments amplifies this fixation, causing it to consistently justify why the output "should" be '12.'

- Finally, in **Stage 6**, the model outputs "I'm confused" and then collapses into a hallucination loop and outputs the word 'Hmm', ultimately being forcefully terminated.

Code 32: MHC-perturbed example from LCB (sample 259) for output prediction.

```
1   def minimumSum(nums: List[int]) -> int:
2       n = len(nums)
3       A = deque()
4       rmin = math.inf
5       for i in range(n - 1, -1, -1):
6           A.appendleft(rmin)
7           rmin = min(rmin, nums[i])
8       lmin = nums[0]
9       ans = math.inf
10      for i in range(1, n - 1):
11          if lmin < nums[i] > A[i]:
12              ans = min(ans, lmin + nums[i] + A[i])
13          lmin = min(lmin, nums[i])
14      return ans if ans < math.inf else -1      # The return value is 12
15  ==================================================================================================
16  """ Suggesting answer: minimumSum(nums = [6, 5, 4, 3, 4, 5]) == -1 """
```

[BACK TO LRMS ANALYSIS]

# G   Limitation

This study is subject to several limitations. Firstly, we mainly focus on algorithms and data structure questions in Python when constructing perturbations and evaluating model reliability using code reasoning. While our methods can be extended to realistic codebases and are directly transferable to other imperative languages, such as C/C++ and Java, they may not generalize to some languages like Prolog, ML, or Assembly because of the code semantics and structure. Secondly, although we evaluate seventeen LLMs and three LRMs, some latest closed-source models (*e.g.*, GPT-4.5, o3, o4-Mini, Gemini-2.5, and Claude-3.7) and open-source models (*e.g.*, Mixtral, Falcon, Vicuna) are not included due to budget constraints. Thirdly, we adopt PASS@1 as the primary evaluation metric. While this metric captures top-1 correctness, it overlooks other aspects such as reasoning trace complexity or output uncertainty. Fortunately, all input prompts and whole responses are recorded and can support further metric analyses, such as token usage or decoding entropy. Lastly, our work does not offer definitive mitigation strategies. Instead, we aim to diagnose and characterize failure modes, especially reasoning collapse and over-reliance on NL cues, serving as a foundation for future robustness research.

