# OpenReview forum: "CodeCrash: Exposing LLM Fragility to Misleading Natural Language in Code Reasoning"
_NeurIPS.cc/2025/Conference — NeurIPS 2025 poster_

### Official Review · Reviewer_RETr · 2025-06-30

**Clarity:** 2
**Significance:** 2
**Originality:** 3
**Rating:** 4
**Confidence:** 4

**Summary:**

Based on my understanding, this paper explores how large language models (LLMs) handle code reasoning by proposing a new benchmark that uses functionally equivalent programs (FEPs). The key idea is that instead of relying on exact string match or canonical reference implementations, which can be brittle, the authors introduce evaluation metrics that assess whether two code snippets behave the same across diverse test inputs. This seems especially relevant for tasks like program repair or completion, where multiple correct solutions may exist.

**Questions:**

See weaknesses.

**Ethical Concerns:**

["NO or VERY MINOR ethics concerns only"]

**Final Justification:**

The authors address most of my concerns.

**Limitations:**

No.

**Quality:**

3

**Strengths And Weaknesses:**

appreciate the motivation, the LLMs often get penalized for producing code that doesn’t match ground truth but still works perfectly fine. So bringing in FEPs to test “true” reasoning ability is quite reasonable. The benchmark design seems thoughtful, with an emphasis on creating clusters of functionally equivalent but syntactically diverse code. Also, the authors make an effort to evaluate a wide range of LLMs, and the observation that some models fall short in generalizing across code variations is a useful insight.

Weaknesses:

I had several concerns while reading. The main one is that the overall novelty feels limited. The idea of using program equivalence (or behavioral testing) for evaluating code generation has been explored in past work. For example, many code generation benchmarks already incorporate execution-based metrics or fuzzing strategies. While this paper pushes that line further with FEP clusters, the contribution feels more incremental than foundational.

Second, the construction of FEP clusters (although clearly non-trivial), is not entirely convincing in terms of scale and coverage. The paper mentions hundreds of examples, but it’s not clear how diverse or comprehensive the test suite is across different programming concepts or complexity levels. It feels like the authors focus more on surface diversity rather than semantic or algorithmic depth, which limits the benchmark’s ability to reveal deeper reasoning gaps.

Finally, the link between “reasoning” and “failing to generalize across FEPs” is a bit hand-wavy. In many cases, LLMs might just rely on memorized patterns or local statistical features, so failing on a FEP variant doesn’t necessarily mean a failure of reasoning. As it could be a lack of exposure. I would have liked to see clearer criteria or ablation studies to isolate reasoning vs memorization effects.

---

> ### Author Rebuttal · Authors · 2025-07-31
>
> We appreciate your thoughtful feedback, particularly regarding our benchmark’s novelty, the scale breadth of the FEP clusters, and the distinction between reasoning and memorization. For your highlighted concerns, we will address them one by one.
>
> ---
> > [W1] The main one is that the overall novelty feels limited. The idea of using program equivalence (or behavioral testing) for evaluating code generation has been explored in past work ... While this paper pushes that line further with FEP clusters, the contribution feels more incremental than foundational.
> ---
>
> Appreciating your thoughtful concerns about novelty, we acknowledge that the idea of using program equivalence for evaluating code generation has been explored before [1,2,3,4], and we indeed consider an incremental exploration based on the previous works [4,5,6]. However, **existing robustness suites and defenses typically study generation-context perturbations, modifying either the natural-language task description [1,2,3] or partial code context within prompts [1,4]**.
>
> In contrast, our work **uniquely focuses on code understanding robustness**, where the model must directly interpret and execute a complete program. While generation tasks focus on the capability of NL to Code, our setting demands semantic analysis of control-flow and data-flow, and explicit understanding of code logic, targeting further tasks like Code Explanation (i.e., **Code to NL capability**). Therefore, existing results on code generation robustness do not diminish our contributions toward the understanding robustness problem.
>
> Additionally, prior robustness evaluations usually use **a single, internally self-consistent perturbation pattern**. For example, modifying code structure (e.g., variable renaming, inserting dead loops) or editing task descriptions (e.g., introducing grammatical errors or misspellings). In contrast, our benchmark introduces **two contradictory yet valid patterns simultaneously** (program logic vs. misleading natural-language cues). This novel setup uncovers a previously undiscussed weakness--**models systematically over-rely on textual cues (not just comments)**, causing them to fail or even collapse under contradictions (§4). **This phenomenon, to the best of our knowledge, has not been previously reported.**
>
> Furthermore, our experiments reveal other surprising findings:
>
> 1. In §4.3.2, **QwQ-32B repeatedly enters a self-verification loop**, characterized by a quadratic and gradual reduction of confidence and eventual reasoning collapse (output of the same tokens). It is triggered simply by one incorrect authoritative comment. Our detailed analysis in §5.3 and Appendix E.6 confirms this collapse is systematic rather than coincidental.
> 2. In §4.4, **misleading textual cues cause models to shortcut reasoning in input prediction tasks**. While input prediction is inherently more challenging than output prediction, its design ironically permits models greater susceptibility to superficial textual shortcuts.
>
> These nuanced insights further illustrate the unique contribution of our work.
>
> [1] ReCode: Robustness Evaluation of Code Generation Models. Shiqi Wang, et. al.
>
> [2] NLPerturbator: Studying the Robustness of Code LLMs to Natural Language Variations. Junkai Chen, et. al.
>
> [3] On Robustness of Prompt-based Semantic Parsing with Large Pre-trained Language Model: An Empirical Study on Codex. Terry Yue Zhuo, et. al.
>
> [4] CCTEST: Testing and Repairing Code Completion Systems. Zongjie Li, et. al.
>
> [5] CRUXEval: A Benchmark for Code Reasoning, Understanding and Execution. Alex Gu, el. al.
>
> [6] LiveCodeBench: Holistic and Contamination Free Evaluation of Large Language Models for Code. Namin Jain, et. al.
>
> ---
> > [W2] Second, the construction of FEP clusters (although clearly non-trivial), is not entirely convincing in terms of scale and coverage.
> ---
>
> Our design goal with the FEP clusters was specifically targeted at evaluating code understanding robustness under two distinct dimensions: structural perturbations and textual perturbations through code execution and input prediction tasks. Thus, our benchmark **emphasizes variability in these robustness-relevant dimensions, rather than aiming at general algorithmic complexity**. Nevertheless, we built upon two complementary datasets: synthetic algorithmic tasks (CRUXEval) and real-world problems (LiveCodeBench), covering multiple complexity levels and common Python constructs. We also provided a balanced and cross-dataset comparison for all experiments.
>
> ---
> > [W3-1] Finally, the link between “reasoning” and “failing to generalize across FEPs” is a bit hand-wavy. In many cases, LLMs might just rely on memorized patterns or local statistical features.
> ---
>
> Regarding your concern about the separation between memorization and reasoning. We consolidate the evidence already in the paper and add a new shuffle ablation that directly addresses this point.
>
> In §4.2 (or Table 1), we design an **RTF perturbation** to rewrite easy branch conditions into indirect and infrequently seen expressions, **eliminating the common lexical pattern**. However, all models maintain a similar level of accuracy in the VAN setting, indicating that they can correctly reason over unfamiliar syntactic forms.
>
> In §4.3, we conduct **CoT prompting** on every model to evaluate the code reasoning robustness again, while prior work shows CoT reduces shortcut behaviour and pattern reliance [7], **model performance still significantly dropped** under ALL, MDC, and MPS perturbations. Thus,  the errors persist after exposing the model's full reasoning trace, **contradicting a pure memorisation or lack-of-exposure hypothesis**.
>
> In §5.2, we add the **"ignore comment" instruction** to the prompt, but the performance of DeepSeek-V3, GPT-4o, and Qwen2.5-72B-Instruct increased, but still **cannot recover their accuracy to VAN-level** under MDC perturbation even using CoT prompting. This shows that the **model does not blindly map comments to answers** (otherwise, the instruction would fully fix the error). The performance gap points to a reasoning failure when NL cues contradict program logic.
>
> [7] Deciphering the Factors Influencing the Efficacy of Chain-of-Thought: Probability, Memorization, and Noisy Reasoning. Akshara Prabhakar, et. al.
>
> ---
> > [W3-2] I would have liked to see clearer criteria or ablation studies to isolate reasoning vs memorization effects.
> ---
>
> To further verify the separation between pattern matching and reasoning, we conduct a **new shuffle ablation** by removing lexical alignment.
>
> In particular, we inject misleading comments specifically to different key AST nodes (which are described in Appendix A5, pp. 20), as we want to enlarge the contradiction between the code logic and misleading comments. **In this new ablation experiment, we shuffle the comment positions**, such as we inject the misleading comments designed for loops into function declarations.
>
> |Model|VAN|MDC|MDC(Shuffled)|
> |:---|:--:|:--:|:---:|
> |GPT-4o|64.5|45.7|54.9|
> |LLaMA-3.3-70B-Ins|48.5|38.7|42.0|
> |Qwen2.5-72B-Ins|54.9|48.0|52.4|
>
> - Observation 1: Accuracy improves when alignment is broken
> - Observation 2: Accuracy is still below the VAN-level.
>
> These observations indicate that the models are not simply doing memorized or lexical pattern-matching. **If they were blindly keyed to comments, shuffling the comments would not help; if they only followed code patterns, accuracy would remain the same because the code logic is unchanged.** Instead, shuffling breaks the line-level alignment between code and comments, leading the model to reason about their inconsistency and place greater trust in the actual code logic, hence the observed improvement, but still below the vanilla baseline.

---

> > ### Comment · Reviewer_RETr · 2025-08-01
> >
> > 1. Could u elaborate on the specific CoT "recipes" or reasoning strategies used in their prompting (e.g., how reasoning steps were decomposed, what structural patterns were employed, and how executable semantics were framed)? While Appendix E provides four case studies, it would be valuable to have a more systematic summary of CoT styles, decomposition strategies, and their mapping to robustness gains or failure modes across perturbation types? (As different recipes may cause high std.)
> >
> > 2. Your perturbation design (especially the AST-based structural variants and textual distractions) is quite effective. However, it seems the current framework abstracts away from structured semantic signals such as AST graphs or control/data-flow graphs, which have been implicitly utilized in prior studies for code robustness and reasoning. It might be beneficial to contextualize your perturbation strategies in relation to this line of work [1,2,3,4,5] perhaps in the related work section, to better position your contributions within the broader landscape.
> >
> > [1] aclanthology.org/2022.coling-1.521
> >
> > [2] arxiv.org/abs/2404.05767
> >
> > [3] www.researchgate.net/publication/373920412_Enhancing_Code_Summarization_with_Graph_Embedding_and_Pre-Trained_Model
> >
> > [4] ieeexplore.ieee.org/abstract/document/10027715
> >
> > [5] dl.acm.org/doi/pdf/10.1145/3636430

---

> > > ### Author Response · Authors · 2025-08-03
> > > **Reply [Q1]: CoT Reasoning**
> > >
> > > Thank you for your comments that can further improve our paper quality.
> > >
> > > Regarding Q1, as mentioned in Appendix C, we follow the single-shot CoT prompt designed by CRUXEval and also employed in LiveCodeBench. Accordingly, all models' CoT traces share a consistent recipe:
> > >
> > > - **Reasoning steps decomposition:** All models **decompose the task by executing the code sequentially**. At each line (or each iteration), they identify the operation and derive its effect on the program state and variable values.
> > >
> > > - **Structural patterns employed:** The reasoning follows the **execution-trace pattern** that the model specifies the current line, the operation executed, and the updated values of key variables.
> > >
> > > - **Executable semantics framing:** At each step (line or iteration), the model computes and states the updated values of variables by producing a **pseudo-execution to demonstrate the actual runtime behavior**. For example:
> > >
> > >   ```
> > >   ...
> > >   for j in range(i + 1, n):
> > >       if nums[j] - nums[j - 1] == delta:
> > >           ans = max(ans, j - i + 1)
> > >           delta = -delta
> > >       else:
> > >       		break
> > >   ...
> > >   ```
> > >
> > >   and the CoT output:
> > >
> > >   ```
> > >   - Start with delta = 1
> > >   - j = 1: nums[1] - nums[0] = 3 - 2 = 1, matches delta, ans = 2, delta = -1
> > >   - j = 2: nums[2] - nums[1] = 4 - 3 = 1, doesn’t match delta (-1), break
> > >   ```

---

> > > > ### Author Response · Authors · 2025-08-03
> > > > **Extended Discussion [Q1]: CoT Behavior under Perturbations**
> > > >
> > > > Actually, we observed that under MDC (misleading comments) and MPS (misleading prints) perturbations, models exhibit different CoT reasoning modes:
> > > >
> > > > - **MDC (misleading comments):**
> > > >
> > > >   - During step-by-step execution, **models typically do not output the comments verbatim. They implicitly incorporate the misleading content into their expectations of the code execution if they were misled, leading to incorrect reasoning.**
> > > >
> > > >   - To make the comments more plausible, we include variable names from the code in the comments. This explains why shuffling the comment positions (shuffle ab) weakens the effect. After shuffling the comment positions, misplaced comments look less credible, so the models have a lower chance of being misled.
> > > >
> > > >   - This also aligns with our findings in §5.1, where the performance drop scales with comment density. Since each line analyzed has a chance of being misled, a higher density will accumulate a stronger misleading effect.
> > > >
> > > > - **MPS (misleading print statements):**
> > > >
> > > >   - **Unlike MDC, models output the print content explicitly as part of their reasoning under MPS-perturbed code.** Since the print statements are inserted near the corresponding code lines, the **models are then misled by their own previously generated content** and reinforce the misleading information. This illustrates a distinct failure mode.
> > > >
> > > >   - Hence, this phenomenon enhanced our stance that models will include any format of NL information embedded in code, not just comments, to analyze. Even if the misleading content is injected through other formats (e.g., `x = "<misleading sentence>"`), the effect may still persist.
> > > >
> > > > To our knowledge, prior work has not explicitly revealed this phenomenon, which underscores one of our key contributions.

---

> > > ### Author Response · Authors · 2025-08-03
> > > **Reply [Q2]: Relation to Structured Semantic Approaches**
> > >
> > > Regarding your second question, we appreciate the recognition of our perturbation effectiveness, particularly the AST-based textual distractions. We would like to clarify and acknowledge that:
> > >
> > > 1. **AST usage:** The **AST is used as a surface-level perturbation mechanism** rather than as a deeper semantic embedding to improve model reasoning.
> > > 2. **Goal:** Our primary goal is to **evaluate the robustness and stress-test reasoning capabilities of LLMs** when subjected to contradictory, albeit superficial, perturbations.
> > >
> > > Precisely because our perturbations are surface-level, **models still induce significant reasoning failures, which highlight an underexplored vulnerability that models over-trust the superficial cues or tend to shortcut their reasoning**.
> > >
> > > =====
> > >
> > > Indeed, **your provided references are valuable and relevant to our work**. They contributed by leveraging deep structural embeddings such as Program Dependency Graphs (PDG) [1], Control Flow Graphs (CFG) [3], or sophisticated AST encodings [2,3] directly embedded into models to enhance internal reasoning and model performance in code-related tasks. For example:
> > >
> > > - Son et al. [1]: Uses PDG embeddings to capture deeper semantic relationships (compared to traditional AST embeddings).
> > >
> > > - Li et al. [3] incorporate Flow-Augmented ASTs by combining CFG and DFD signals.
> > >
> > > - CSA-Trans [2] and StructCoder [5] incorporate the structural graph embeddings into the transformer architectures to improve the intrinsic reasoning capacity.
> > >
> > > - Interestingly, Pei et al. [4] introduced contrastive code-comment pre-training to enhance deep semantic understanding through NL and structural code embedding.
> > >
> > >   **This may inadvertently amplify the vulnerability discovered by our work**, as it will reinforce the model's reliance on natural language comments for program comprehension.
> > >
> > > Since our work aims at exploring the vulnerabilities of existing models when confronted with simpler and externally applied perturbations. In this sense, structure-aware modeling [1-5] and our perturbation-based evaluation are complementary. **While the former aims to improve capability, our work aims to highlight and diagnose the vulnerabilities.**
> > >
> > > **We have included these valuable references and discussions in our related work section to more clearly position our contributions** within the broader landscape of code reasoning robustness research.
> > >
> > > [1] Boosting Code Summarization by Embedding Code Structures. Jikyeong Son et al.
> > >
> > > [2] CSA-Trans: Code Structure Aware Transformer for AST. Saeyoon Oh and Shin Yoo.
> > >
> > > [3] Enhancing Code Summarization with Graph Embedding and Pre-Trained Model. Lixuan Li et al.
> > >
> > > [4] Contrastive Code-Comment Pre-training. Xiaohuan Pei et al.
> > >
> > > [5] StructCoder: Structure-Aware Transformer for Code Generation. Sindhu Tipirneni et al.

---

> > > > ### Comment · Reviewer_RETr · 2025-08-03
> > > >
> > > > Thank you for your effort. I still strongly suggest adding **main inllustration Figure** to elabrate your propoed perturbed code perturbation strategies (*Section 3.3.1 - 3.3.2*).  When I read your paper, It is quite difficult to follow the newly introduced terminology, as several concepts are only briefly mentioned and then referred to using abbreviations. This significantly affects the overall readability. I recommend improving the presentation to make your contributions clearer. While the paper does explain what has been done, it tends to obscure the core ideas through implicit in Appendix, abbreviations, and newly coined terms. Altogether, these issues give the impression that the paper seems to be a draft that not ready to publish, potentially due to authours' time constraints.
> > > >
> > > > Nonetheless, I sincerely appreciate your thoughtful rebuttal and the effort you have made in responding. I raised my score based on the rebuttal efforts.

---

> > > > > ### Author Response · Authors · 2025-08-04
> > > > >
> > > > > Thank you for your constructive feedback. We agree that improving readability is crucial, so we have revised our work:
> > > > >
> > > > > 1. **Main illustration figure:** We moved the original Figure 3 from Appendix A to §3 with focusing on the modified parts and highlighting our AST operations (search → replace, reformat, insert).
> > > > > 2. **Streamlined results presentation:** Since we put the CoT and Input Prediction results tables from the Appendix back in the main text, we shortened the result reports and expected findings, and emphasized the novel findings (e.g., the CoT reasoning under perturbed-code *(Extended Discussion)* and vulnerabilities of input prediction).
> > > > > 3. **Improved coherence in analysis:** We combined Sections 5.3 and 4.3.2, such that readers can read our continuous analysis and discussion of QwQ-32B reasoning collapse.
> > > > >
> > > > > Thank you very much for your time and effort in reviewing our paper. Your suggestions have greatly improved the clarity and quality of our work.

---

### Official Review · Reviewer_dLhF · 2025-07-03

**Clarity:** 3
**Significance:** 2
**Originality:** 2
**Rating:** 4
**Confidence:** 4

**Summary:**

The paper introduces CodeCrash, a benchmark/framework for evaluating Code LLMs under perturbed settings. Starting from LiveCodeBench and CruxEval, the authors design code semantics-preserving perturbations that either modify code structures or textual descriptions (comments, variable names) to obtain perturbed task problems. The authors present systematic evaluations across settings and show that non-reasoning models degrade performance under the perturbed settings. Reasoning language models are able to not only achieve very high performance under the non-perturbed setting but also be more robust to perturbations. Finally, the authors present detailed discussions around three settings — varying NL comments density in textual perturbations, prompting to ignore misleading comments and identifying self-reflection loops in reasoning models.

**Questions:**

1. How do the perturbations discussed in this paper relate to those in recent code perturbation works such as CodeFort or ReCode? Are there new findings compared to those from these works?
2. On reasoning models evaluations, do Section 5.3 findings hold even for the VAN setting? In other words, is the self-reflection collapse surely due to the perturbation?
3. Is it possible to identify adversarial scenarios in real-world code? For e.g., code with arbitrary comments, missing imports, unintelligible variable names, etc.? This would ground the research in more practical settings.
4. Are there recommendations for improving robustness under various perturbation settings?

**Ethical Concerns:**

["NO or VERY MINOR ethics concerns only"]

**Final Justification:**

The authors have highlighted that their main goal is to stress-test LLMs under adversarial scenarios concerning code reasoning. From that perspective, their work builds on existing robustness work before code reasoning was prevalent. As such, this paper presents timely findings for robustness of code reasoning.

**Limitations:**

No limitations are discussed in the paper. I recommend the authors to address the above.

There are no ethical considerations around this work however.

**Quality:**

2

**Strengths And Weaknesses:**

**Strengths**

The paper discusses a comprehensive set of perturbations which reveals gaps in LLMs’ understanding of code. Further, the paper supports the superiority of reasoning language models and their (current) tendency to overthink or enter into self-reflection loops.

**Weaknesses**

- **Insufficient contributions relative to literature** Evaluating code LLMs under perturbed settings such as those described in the paper is not new. The paper only presents findings that newer small models continue to suffer from this problem while larger or reasoning models tend to suffer less. This is also, in my opinion, largely known to the community [4]. Further, the benchmark is merely derived from LiveCodeBench and CruxEval by introducing perturbations that are unrealistic or not representative of real world use cases (such as introducing misleading hints). From an analysis perspective, I understand that this is a controlled experimental setting but the analysis itself is also incomplete in my opinion. Some other works around code robustness such as CodeFort [1] attempt to make models more robust, while the current paper over emphasizes on a largely known problem without providing possible fixes. Finally, the finding that reasoning models may overthink is not a new finding either [2, 3].
- **Paper writing** Given the above, I feel the paper is overly verbose in what needs to be conveyed. Most of the paper writing feels like verbalizing the results in the tables. The paper space can better be utilized by discussing more insightful experiments such as identifying adversarial patterns in real-world code (from GitHub, etc.) and how models behave under such scenarios or how to improve robustness for both reasoning and non-reasoning models. The discussions in Section 5, though insightful, are not significant enough, in my opinion, to merit a full-paper at NeurIPS.
- **Misc.**
    - Line 78: in → on
    - In the present form, the main text makes several references to core results in the Appendix. I feel that the main text should be self-sufficient to the reader and any references to Appendix material should be optional for the reader to look up; this is not possible in the current write-up where I cannot follow some discussions (e.g., Section 4.3.1) without going to the Appendix.

[1] CodeFort: Robust Training for Code Generation Models. https://arxiv.org/pdf/2405.01567

[2] Do NOT Think That Much for 2+3=? On the Overthinking of o1-Like LLMs. https://arxiv.org/abs/2412.21187v2

[3] Thinking LLMs: General Instruction Following with Thought Generation. https://arxiv.org/abs/2410.10630v1

[4] ReCode: Robustness Evaluation of Code Generation Models. https://arxiv.org/pdf/2212.10264

---

> ### Author Rebuttal · Authors · 2025-07-31
>
> Thank you for your hard work reviewing our paper! We appreciate your comments for raising important points on novelty, realism of perturbations, the interpretation of reasoning versus memorization, and providing your feedback on writing clarity. Below we address these concerns in detail.
>
> ---
> > [W1] The paper only presents findings that newer small models continue to suffer from this problem while larger or reasoning models tend to suffer less. This is also, in my opinion, largely known to the community [4].
>
> > [Q1] How do the perturbations discussed in this paper relate to those in recent code perturbation works such as CodeFort or ReCode? Are there new findings compared to those from these works?
> ---
>
> Existing robustness suites and defenses study **generation-context perturbations**, edit the NL task description [1,2,3], or partial code inside the prompt [1,4] that merely serves as context for a code completion task. In contrast, our work is the first benchmark to investigate the **direct code understanding robustness**, where the model must interpret and execute a complete program. This setting demands **semantic analysis of control-flow and data-flow, and the ability to follow the true code logic**.
>
> Code generation/completion and code understanding are related, but they test **two fundamentally different capabilities**. Robustness in code generation primarily measures resilience against prompt variations (NL to Code), while **robustness in code understanding assesses whether a model can correctly analyze and explain the code logic under perturbed conditions (Code to NL)**. Therefore, evidence from code generation robustness does not invalidate our findings on code understanding robustness.
>
> Furthermore, prior works perturb only a **single self-consistent pattern** (e.g., variable renames, death loop inserts, which focus on code logic); **our work superimposes two contradictory yet valid patterns (i.e., program logic vs. misleading comments)**. This novel setup exposes a weakness that models systematically over-trust textual cues. **Two valid but contradictory patterns exist simultaneously, causing the model to fail to reconcile and collapse, which has not been seen in earlier works**.
>
> [1] ReCode: Robustness Evaluation of Code Generation Models. Shiqi Wang, et. al.
>
> [2] NLPerturbator: Studying the Robustness of Code LLMs to Natural Language Variations. Junkai Chen, et. al.
>
> [3] On Robustness of Prompt-based Semantic Parsing with Large Pre-trained Language Model: An Empirical Study on Codex. Terry Yue Zhuo, et. al.
>
> [4] CCTEST: Testing and Repairing Code Completion Systems. Zongjie Li, et. al.
>
> ---
> > [W2] Further, the benchmark is merely derived from LiveCodeBench and CruxEval by introducing perturbations that are unrealistic or not representative of real world use cases.
>
> > [W4] The paper space can better be utilized by discussing more insightful experiments such as identifying adversarial patterns in real-world code (from GitHub, etc.) and how models behave under such scenarios or how to improve robustness for both reasoning and non-reasoning models.
> ---
>
> We fully understand your concern that the perturbations introduced in our benchmark may appear extreme or not directly representative of typical real-world scenarios. We would first like to clarify that our primary goal is explicitly stress-testing LLMs' **capability boundaries in code reasoning, and reversing the NL dependencies of LLMs**. Our perturbations are carefully designed to reveal robustness issues that would remain hidden in traditional evaluation.
>
> To address this realism, we design the **MPS perturbation that injects the same misleading phrases as print statements, which should never influence the output**. However, it negatively impacts model code reasoning, revealing that the **model would reference any NL cues within the code**, which **highlights the unreliability of the model in code reasoning**.
>
> Besides, we included a density sweep experiment in §5.1; we can see that the **Pass@1 accuracy falls linearly**, which shows the **vulnerability is systematic, not a product of extreme cases**. The sparse and accidental hints or comments could be frequently found in real repositories (e.g., CPython GitHub issue 136764).
>
> ---
> > [W3] Finally, the finding that reasoning models may overthink is not a new finding either [2, 3].
>
> > [Q2] On reasoning models evaluations, do Section 5.3 findings hold even for the VAN setting? In other words, is the self-reflection collapse surely due to the perturbation?
> ---
>
> We agree that overthinking has already been discovered in prior works, and we also realize that models will overthink and self-reflect under non-ideal coding situations. However, **our study uncovers a qualitatively different failure mode --- reasoning collapse**. This phenomenon **solely exists in MHC perturbation and happened 4 times, which is not unique and coincidental**. Therefore, answering Q2, reasoning collapse does not hold for the VAN setting, and it is **surely due to the MHC perturbation**. We respectfully note that we have already provided a detailed analysis of this phenomenon in §4.3.2 and §5.3 (and Case study 6). We have manually segmented the collapsed thinking into six stages of gradual confidence erosion in one case in the Appendix (pp. 39-40). Please feel free to see how the model gradually collapses!
>
> ---
> > [Q3] Is it possible to identify adversarial scenarios in real-world code? For e.g., code with arbitrary comments, missing imports, unintelligible variable names, etc.? This would ground the research in more practical settings.
> ---
>
> We implemented a lightweight **refactoring agent** that uses LLM itself (no AST tools but allows CoT thinking) to **(i) delete misleading comments, (ii) rename unintelligible identifiers, (iii) add missing imports, and (iv) strip dead code**. Hence, if the performance increased on the refactored code, it means the model can identify those issues.
>
> Here are the results on LiveCodeBench:
>
> |Setting|VAN|ALL|MDC|MPS|
> |:----|:---:|:---:|:---:|:---:|
> |GPT-4o (Original)|64.5|46.2|45.7|49.6|
> |GPT-4o (After Refactoring)|-|55.3|41.1|54.9|
>
> The performance under the ALL-perturbation setting improved, which means that **the model can identify and neutralize some code structural perturbations**, but it cannot recover to the ideal-level code.
>
> Surprisingly, in the MDC setting, **the model is still impacted by the misleading comments even when we explicitly remove instructions**, which reinforces our stance and highlights the inherent fragility.
>
> ---
> > [Q4] Are there recommendations for improving robustness under various perturbation settings?
> ---
>
> In our paper (§5.2), we confirmed that **prompt engineering** is a simple yet effective baseline strategy for enhancing robustness, but it cannot fully eliminate the reliance. During the rebuttal period, we explored the solution from **MCP AI Agents (mcp-clean-code)**:
>
>   |Setting|VAN|ALL|MDC|MPS|
>   |:----|:---:|:---:|:---:|:---:|
>   |Direct Inference|73.8|48.6|66.7|48.9|
>   |Direct Inference (w/ MCP)|89.8|79.7|75.6|75.6|
>   |CoT Inference|89.8|78.5|86.2|81.4|
>
> While initially promising, MCP's inherent design targets code generation/planning, which is not suitable for code reasoning. Surprisingly, MDC and MPS underperformed basic CoT prompting, suggesting the **need for more reasoning-specific architectures**.
>
> Due to the time limit, we were unable to empirically evaluate deeper architectural modifications, such as syntax-aware encoders and tree-sitter MCP. Despite our mixed results, we believe our current explorations can demonstrate the importance of addressing NL dependency.
>
> ---
> > [Misc 1] Line 78: in → on
> ---
>
> Thank you for pointing that out. We have corrected it.
>
> ---
> > [Misc 2] In the present form, the main text makes several references to core results in the Appendix. I feel that the main text should be self-sufficient to the reader and any references to Appendix material should be optional for the reader to look up; this is not possible in the current write-up where I cannot follow some discussions (e.g., Section 4.3.1) without going to the Appendix.
> ---
>
> Thank you for pointing out this issue. We apologize for the inconvenience during your reading. Owing to the page limit, we originally moved the result tables for §4.3.1 and §4.4 to the Appendix, which indeed forced readers to flip pages.
>
> We **have already restored both tables to the main text** in our latest version. To recover the necessary 0.5 page, we've done the following revision:
> 1. Since Appendix A already provides low-level descriptions of each perturbation, we use two concise, high-level paragraphs to summarize the code-structural and textual-distraction perturbations, respectively, rather than using 8 separate paragraphs.
> 2. We moved the dataset descriptions (§3.1) to Appendix A, while adding a one-sentence summary plus a navigation in §3.2 (execution pipeline).
>      These changes keep the paper within the 9-page limit while readers can grasp the methodology at a high level and see all core results without flipping to the Appendix.
>
> ---
> > [Limitation 1] No limitations are discussed in the paper.
> ---
>
> There is an oversight on our side that the limitation section appears in Appendix F (p. 40), and we realise that we did not mention this Appendix section in the main text. We have already included a sentence with explicit forward and backward links to the limitation section in the discussion section, so that readers who want details can navigate effortlessly.

---

> > ### Author Response · Authors · 2025-08-06
> >
> > Dear Reviewer dLhF,
> >
> > Thank you once again for your valuable comments! As the discussion phase is approaching its end, we would like to kindly confirm whether we have sufficiently addressed all of your concerns (or at least some of them). Should there be any remaining questions requiring further clarification, please do not hesitate to let us know. If you are satisfied with our responses, we would greatly appreciate your consideration in adjusting the evaluation scores accordingly.
> >
> > We sincerely look forward to your feedback.

---

### Official Review · Reviewer_5hDL · 2025-07-03

**Clarity:** 3
**Significance:** 2
**Originality:** 2
**Rating:** 4
**Confidence:** 3

**Summary:**

This paper investigates the robustness of LLMs in coding reasoning tasks under perturbations by proposing a new benchmark called CodeCrash. The evaluation on 17 LLMs demonstrates that perturbation can cause significant performance degradation. This study offers insights into the limits of current benchmarks on robust code understanding.

**Questions:**

1. Could the framework be extended to multi programming language settings? Is there any pipeline to do that?

**Ethical Concerns:**

["NO or VERY MINOR ethics concerns only"]

**Final Justification:**

My concerns has been addressed. I will keep my positive score.

**Limitations:**

Yes

**Quality:**

3

**Strengths And Weaknesses:**

Strengths:
1. The manuscript is well written and easy to follow
2. The proposed benchmark is meticulously curated, with clear definitions of each perturbation type.
3. The experimental analysis is thorough and insightful, offering both broad comparisons and fine-grained breakdowns.

Weaknesses:
1. Although comprehensive, the experimental results align closely with expectations given the inherent difficulty of the original benchmarks (e.g., livecodebench), offering limited surprise for the reader.
2. As a comprehensive benchmark, while most main stream and state-of-the-art non reasoning models are covered, the number of reasoning models covered in the benchmark is still limited. Only three are evaluated.
3. The benchmark relies heavily on questions drawn from existing datasets, which constrain the technical contributions of the paper and real-world applicability.

---

> ### Author Rebuttal · Authors · 2025-07-31
>
> Thank you for your hard work reviewing! We appreciate your recognition of the strengths of our work. We will address your concerns one by one.
>
> ---
> > [W1] Although comprehensive, the experimental results align closely with expectations given the inherent difficulty of the original benchmarks (e.g., livecodebench), offering limited surprise for the reader.
> ---
>
> We agree that several findings match the intuitive expectations. For example, the performance drop under perturbations (§4.2), the mitigating effect of CoT (§4.3.1), and the outstanding results of reasoning models (§4.3.2). At the same time, we would like to highlight **two unexpected findings** that add novelty beyond these trends:
> 1. **Reasoning collapse in QwQ-32B**. The model entered a self-reflection loop when faced with a plausible but incorrect hint, leading to a catastrophic cognitive-dissonance failure that we did not anticipate.
> 2. **Performance increase in input-prediction under MDC/MPS (§4.4)**. Misleading strings accidentally supply the model with shortcuts, **exposing a potential flaw in input-prediction tasks** and offering new evidence of unreliable code understanding.
>
> ---
> > [W2] As a comprehensive benchmark, while most main stream and state-of-the-art non reasoning models are covered, the number of reasoning models covered in the benchmark is still limited. Only three are evaluated.
> ---
>
> We acknowledge that only three reasoning models were included is limited. However, the evaluation cost is substantial, testing these three models across four perturbation types (and a baseline) on 1279 problems already **exceeded $100 USD in API usage**. We prioritized breadth in non-reasoning models while still ensuring representative reasoning-model coverage.
>
> ---
> > [W3] The benchmark relies heavily on questions drawn from existing datasets, which constrain the technical contributions of the paper and real-world applicability.
> ---
>
> Since **our primary goal is not to introduce a brand-new dataset, but rather to stress-test LLMs in abnormal and non-ideal scenarios** and highlight concrete robustness issues in code understanding. Using established benchmarks like LiveCodeBench and CruxEval ensures that our findings are grounded in well-validated tasks.
>
> We fully understand your concern that those comment-based perturbations are an extreme and unrealistic case. To address this realism, we design the **MPS perturbation that injects the same misleading phrases as print statements, which should never influence the output**. However, it negatively impacts model code reasoning, revealing that the **model would reference any NL cues within the code, which highlights the unreliability of the model in code reasoning**.
>
> Besides, we included a density sweep experiment in §5.1; we can see that the **Pass@1 accuracy falls linearly**, which shows the **vulnerability is systematic, not a product of extreme cases**. The sparse and accidental hints or comments could be frequently found in real repositories (e.g., CPython GitHub issue 136764).
>
> ---
> > [Q1-1] Could the framework be extended to multi programming language settings?
> ---
>
> To evaluate cross-language generalization, we converted LiveCodeBench into JavaScript (similar complexity to Python). JavaScript Results (Python in parentheses):
>
> |Model|VAN|ALL|MDC|MPS|MHC|
> |:----|:---:|:---:|:---:|:---:|:---:|
> |GPT-4o (Direct)|56.0 (64.5)|-17.5 (-18.3)|-6.5 (-18.8)|-7.1 (-15.7)|-7.9 (-17.2)|
> |GPT-4o (CoT)|94.8 (94.4)|-11.7 (-7.9)|-7.3 (-6.9)|-3.3 (-6.7)|-1.0 (+0.4)|
> |Qwen2.5-72B-Instruct (Direct)|54.1 (54.9)|-17.8 (-13.9)|-9.2 (-7.0)|-10.4 (-14.4)|-24.8 (-22.3)|
> |Qwen2.5-72B-Instruct (CoT)|86.0 (90.4)|-16.9 (-17.1)|-2.7 (-5.0)|-6.7 (-14.2)|-9.0 (-7.7)|
>
> 1. Baseline accuracies are **comparable across the two languages**.
> 2. **Robustness patterns remain consistent**, with notable drops under ALL, MDC, MPS, and MHC perturbations.
> 3. The **mitigating effect of CoT persists** across languages, especially for GPT-4o under MHC.
>
> Therefore, CodeCrash exposes language-agnostic vulnerabilities, not artifacts specific to Python. The identified weaknesses generalize across scripting languages. We have added this result to the appendix in our paper.
>
> ---
> > [Q1-2] Is there any pipeline to do that?
> ---
>
> We have provided the detailed implementation description guidance for Python code in Appendix A. For the code structural perturbations, we have already provided syntax-preserved mapping templates. For the misleading comments, we saved them in "prompt/misleading_comments.py" in the supplementary code, and offered a function to generate those misleading comments.
>
> For the perturbation process, we solely use the AST library to conduct the structural modification and the NL cues injection, which **can be easily transferable to other languages**, such as JavaScript using the **tree_sitter library** and Java using the **javalang library** for handling the JavaScript and Java code snippets. Because the perturbation logic is parser-agnostic and templated, extending CodeCrash to a new language is a plug-and-play operation: (1) import the grammar, (2) wire the formatter, and (3) reuse the existing passes.

---

> > ### Author Response · Authors · 2025-08-06
> >
> > Dear Reviewer 5hDL,
> >
> > Thank you once again for your valuable comments! As the discussion phase is approaching its end, we would like to kindly confirm whether we have sufficiently addressed all of your concerns (or at least some of them). Should there be any remaining questions requiring further clarification, please do not hesitate to let us know. If you are satisfied with our responses, we would greatly appreciate your consideration in adjusting the evaluation scores accordingly.
> >
> > We sincerely look forward to your feedback.

---

> > > ### Comment · Reviewer_5hDL · 2025-08-06
> > > **Response to Rebuttal**
> > >
> > > Thank you for the detailed reply. My concerns has been addressed. I will keep my positive score. Good luck on your submission!

---

> > > > ### Author Response · Authors · 2025-08-06
> > > >
> > > > Thank you very much for your recognition of the value of our work!

---

### Official Review · Reviewer_wYbH · 2025-07-03

**Clarity:** 3
**Significance:** 2
**Originality:** 2
**Rating:** 4
**Confidence:** 3

**Summary:**

This paper introduces CODECRASH, a comprehensive benchmark for evaluating the code reasoning robustness of LLMs. The authors systematically assess 17 LLMs and 3 Large Reasoning Models (LRMs) by applying a series of structural and textual perturbations to code snippets from the CRUXEVAL and LIVECODEBENCH datasets. The perturbations are designed to challenge models by altering code structure without changing its logic and by injecting misleading natural language cues.

**Questions:**

- Cross-Language Generalization: How does CODECRASH perform on non-Python languages, and do LLMs exhibit similar robustness patterns?
- Long-Tail Code Handling: Can LLMs maintain accuracy on extremely complex code (e.g., nested macros, low-level optimizations) under perturbations?
- Mitigation Strategies: Are there architectural modifications (e.g., syntax-aware encoders) that can reduce NL dependency in code reasoning?
- Dynamic Perturbations: How do time-varying perturbations (e.g., evolving codebases) impact LLM performance over continuous deployment?

**Ethical Concerns:**

["NO or VERY MINOR ethics concerns only", "Major Concern: Improper research involving human subjects"]

**Final Justification:**

After the revision. I raised the ratings to "boardline accept".

**Limitations:**

refer to weakness

**Quality:**

3

**Strengths And Weaknesses:**

Strengths
The paper trying to addresses a highly significant and timely problem. As LLMs become integrated into daily software development workflows, understanding their reliability and failure modes under non-ideal conditions is critical for safe and effective deployment. This work moves beyond standard code generation benchmarks to probe the more fundamental capability of code reasoning.
The CODECRASH benchmark is an original and valuable contribution. The perturbation strategies are systematic, well-motivated by real-world code "messiness," and thoughtfully designed to target different aspects of a model's comprehension abilities. ex reasoning mechanisms.
The paper is well-written and easy to follow. The methodology is clearly explained with illustrative examples, and the results are presented effectively through comprehensive tables and figures. The logical flow from motivation to methodology to results and discussion is clear and convincing.
Weaknesses
Despite the paper's strengths in its empirical execution, there are several conceptual weaknesses in the interpretation of the results that limit its overall impact.

The paper's central claim is that LLMs rely on NL cues instead of "true" logical reasoning. However, the methodology may not be sufficient to substantiate this strong claim. By removing semantic variable names and replacing them with generic placeholders like f or Var_1, the test confirms that models are brittle to the removal of one type of pattern (semantic). It does not, however, prove that the models' subsequent success is due to "reasoning." It is equally plausible that the models are simply falling back on another form of sophisticated pattern matching, this time based on structural patterns learned from countless examples of algorithmic code that use such generic naming conventions. The work effectively swaps one pattern-matching test for another, leaving the fundamental question of whether these models can "reason" in a human-like, abstract way still open.

The paper frames the models' reliance on comments and hints as a fundamental flaw. This perspective is debatable. In real-world software engineering, relying on high-quality documentation and comments is a feature of an expert developer, not a bug. The core issue revealed by CODECRASH is not that LLMs use NL cues, but that they cannot critically assess the trustworthiness of those cues. The benchmark's adversarial setup, which intentionally injects false information, conflates "using NL cues" with "naively trusting all NL cues." A more insightful framing would explore this as a failure of critical evaluation or a form of sycophancy, rather than a failure of reasoning itself.

The discovery of "reasoning collapse" in QwQ-32B is a fantastic finding. However, the analysis does not go deep enough in explaining why this happens in the context of broader LLM alignment research. The phenomenon can be interpreted as a form of "cognitive dissonance," where the model is unable to resolve a conflict between its own step-by-step logical derivation and an external, authoritative-looking hint. This catastrophic failure to reconcile conflicting information is deeply related to known LLM issues like sycophancy (over-trusting the prompt) and over-thinking. The paper identifies the symptom but misses the opportunity to connect it to the underlying disease, thus limiting the theoretical insight.

The CODECRASH benchmark is a static, one-shot evaluation. This does not reflect the dynamic, interactive nature of how developers use AI assistants. In practice, a model that produces a wrong answer based on a misleading comment would likely be corrected by the developer in a feedback loop. The paper's conclusions about model "reliability" are based on an autonomous agent framing that may not be generalizable to the primary, human-in-the-loop use case. This limits the practical implications of the findings.

---

> ### Author Rebuttal · Authors · 2025-07-31
>
> Thank you for your hard work reviewing! We appreciate that you provided us with such thoughtful feedback. Your insights greatly enrich our work and will help us clarify and strengthen the interpretation of our findings. For your highlighted concerns, we will address them one by one.
>
> ---
> > [W1] The paper's central claim is that LLMs rely on NL cues instead of "true" logical reasoning. However, the methodology may not be sufficient to substantiate this strong claim.
> ---
>
> We would like to clarify that our goal is **not to equate success under REN** with "true or human-like reasoning". In §4.2.1, the performance degradation **solely reveals that models are willing to leverage semantic cues to assist their reasoning**. Based on this property, we further extend our evaluation to misleading NL information.
>
> To further disentangle pattern-matching from reasoning effects, we point you to our case analyses (Cases 2 to 4) on pages 34 - 36. **Especially in case 4, Claude-3.5-Sonnet step-by-step trace reaches the correct understanding, but finally adopts the misleading comment**. The final answer is overridden, demonstrating **a reasoning override by a misleading hint, not via alternate structural patterns**.
>
> ---
> > [W2] The paper frames the models' reliance on comments and hints as a fundamental flaw.
> ---
>
> We agree that high-quality comments are a feature of an expert developer. However, **our goal is not to penalize or claim that "using comments" is inherently bad**. Unlike other tasks, coding has a deterministic ground truth defined by the program's logic, so **strong models should primarily adhere to the true logic of the code**, treating comments or other NL cues as auxiliary signals, rather than as overriding evidence when there are conflicts. In our case studies (Case 2-4), we found that models were simply misled by NL cues to override their inference, which is **undoubtedly a failure in code understanding**.
>
> We understand that models may be trained to treat comments as authoritative instructions, so we introduced **MPS**, which injects the same misleading phrases as print strings that are unrelated to the return value. However, exhibited similar degradation under MPS and MDC, indicating **an underlying tendency to leverage any NL cues from the code to help them understand the code**. We admit that our paper may overhighlight the "reasoning failure"; we will revise our wording to emphasize this phenomenon as a failure of critical evaluation.
>
> ---
> > [W3] The paper identifies the symptom but misses the opportunity to connect it to the underlying disease, thus limiting the theoretical insight.
> ---
>
> We respectfully note that we have already provided a detailed analysis of this phenomenon in §4.3.2 and §5.3 (and Case study 6). We uncover that the **QwQ-32B collapsed four times, exclusively under the MHC perturbation**, so it is **not unique and a coincidence**. In pp. 39-40, we manually segmented the collapsed thinking into six stages of gradual confidence erosion. Our analysis in §5.3 quantifies the decline of "confidence", showing a quadratic increase in confusing tokens, and attributing it to the **contradiction between authoritative yet misleading hints and self-reasoning**.
>
> To strengthen the connection with broader alignment research, we have revised the discussion §5.3 to explicitly frame reasoning collapse as a form of "cognitive dissonance" with our concluded triggering conditions.
>
> ---
> >  [Q1] Cross-Language Generalization: How does CODECRASH perform on non-Python languages, and do LLMs exhibit similar robustness patterns?
> ---
>
> To evaluate cross-language generalization, we converted LiveCodeBench into JavaScript (similar complexity to Python). JavaScript Results (Python in parentheses):
>
> |Model|VAN|ALL|MDC|MPS|MHC|
> |:----|:---:|:---:|:---:|:---:|:---:|
> |GPT-4o (Direct)|56.0 (64.5)|-17.5 (-18.3)|-6.5 (-18.8)|-7.1 (-15.7)|-7.9 (-17.2)|
> |GPT-4o (CoT)|94.8 (94.4)|-11.7 (-7.9)|-7.3 (-6.9)|-3.3 (-6.7)|-1.0 (+0.4)|
> |Qwen2.5-72B-Instruct (Direct)|54.1 (54.9)|-17.8 (-13.9)|-9.2 (-7.0)|-10.4 (-14.4)|-24.8 (-22.3)|
> |Qwen2.5-72B-Instruct (CoT)|86.0 (90.4)|-16.9 (-17.1)|-2.7 (-5.0)|-6.7 (-14.2)|-9.0 (-7.7)|
>
> 1. Baseline accuracies are **comparable across the two languages**.
> 2. **Robustness patterns remain consistent**, with notable drops under ALL, MDC, MPS, and MHC perturbations.
> 3. The **mitigating effect of CoT persists** across languages, especially for GPT-4o under MHC.
>
> Therefore, CodeCrash exposes language-agnostic vulnerabilities, not artifacts specific to Python. The identified weaknesses generalize across scripting languages. We have added this result to the appendix in our paper.
>
> ---
> > [Q2] Long-Tail Code Handling: Can LLMs maintain accuracy on extremely complex code (e.g., nested macros, low-level optimizations) under perturbations?
> ---
>
> To explore the long-tail complexity, we use an industrial-level JavaScript Obfuscator to obfuscate the code in LiveCodeBench (JavaScript version) in **3 levels of obfuscation**:
>
> - **Lite**: Control flow flattening + renaming variables to "mangled"
> - **Med**: Lite + renaming variables to "hexadecimal" + dead code injection
> - **Hard**: Med + compact code + string array transformations + unicode escape sequences
>
> |Setting|Pass@1|
> |:----|:----:|
> |VAN|94.8|
> |ALL|83.1|
> |JS-Obfuscator-Lite|93.5|
> |JS-Obfuscator-Med|87.1|
> |JS-Obfuscator-Hard|80.6|
>
> Notably, **our designed perturbations outperform the medium obfuscation** setting and are comparable to the hard level, which underscores the effectiveness of our perturbations. Our designed semantic and structural distractions (which can be easily identified by humans) can **disrupt model reasoning even more effectively than standard code obfuscation methods aimed at human readability**.
>
> ---
> > [Q3] Mitigation Strategies: Are there architectural modifications (e.g., syntax-aware encoders) that can reduce NL dependency in code reasoning?
> ---
>
> We appreciate the reviewer’s insightful suggestion, such as syntax-aware encoders, to mitigate the issues.
> In our paper (§5.2), we confirmed that **prompt engineering is a simple yet effective baseline strategy for enhancing robustness**, but it cannot fully eliminate the reliance. During the rebuttal period, we explored several mitigation strategies within our current architecture scope:
>
> - **Multi-agent (Refactoring agent)**: We implemented a lightweight agent that uses LLM itself (no AST tools but allows CoT thinking) to refactor the perturbed code. Then, ask for the same agent to infer.
>
>   |Setting|VAN|ALL|MDC|MPS|
>   |:----|:---:|:---:|:---:|:---:|
>   |GPT-4o (Original)|64.5|46.2|45.7|49.6|
>   |GPT-4o (After Refactoring)|-|55.3|41.1|54.9|
>
>   The performance under the ALL-perturbation setting improved, which means that **the model can identify and neutralize some code structural perturbations**, but it cannot recover to the ideal-level code. Surprisingly, in the MDC setting, **the model is still impacted by the misleading comments even when we explicitly remove instructions**, which reinforces our stance and highlights the inherent fragility.
>
> - **MCP AI Agents (mcp-clean-code)**:
>
>   ---
>   > [W4] This does not reflect the dynamic, interactive nature of how developers use AI assistants.
>   ---
>
>   Inspired by the weakness mentioned by the reviewer, we tried to explore the solution from MCP AI Agents.
>
>   |Setting|VAN|ALL|MDC|MPS|
>   |:----|:---:|:---:|:---:|:---:|
>   |Direct Inference|73.8|48.6|66.7|48.9|
>   |Direct Inference (w/ MCP)|89.8|79.7|75.6|75.6|
>   |CoT Inference|89.8|78.5|86.2|81.4|
>
>   While initially promising, MCP's inherent design targets code generation/planning, which is not suitable for code reasoning. Surprisingly, MDC and MPS underperformed basic CoT prompting, **suggesting the need for more reasoning-specific architectures**.
>
> Due to the time limit, we were unable to empirically evaluate deeper architectural modifications, such as syntax-aware encoders and tree-sitter MCP. Despite our mixed results, we believe our current explorations can demonstrate the importance of addressing NL dependency.
>
> ---
> > [Q4] Dynamic Perturbations: How do time-varying perturbations (e.g., evolving codebases) impact LLM performance over continuous deployment?
> ---
>
> We acknowledge that CodeCrash is a one-shot benchmark, whereas real-world development is iterative. Our choice was deliberate: a static setting allows us to cleanly and directly assess a model’s intrinsic robustness to misleading or perturbed code, providing the community with a worst-case first impression.
>
> We agree that robustness evaluation under evolving codebases is highly valuable. However, such a study requires a versioned dataset with realistic insertions/deletions and a protocol for repeated model interaction, which is beyond the scope of this work. Recent efforts, such as EVALOOP [1], propose a closed-loop robustness benchmark for code LLMs, showing that performance consistently declines under repeated modifications. We appreciate the reviewer's suggestion, and we view this as a natural and important extension of our work.
>
> [1] EVALOOP: Assessing LLM Robustness in Programming from a Self-consistency Perspective. Fang et. al.

---

> > ### Author Response · Authors · 2025-08-06
> >
> > Dear Reviewer wYbH,
> >
> > Thank you once again for your valuable comments! As the discussion phase is approaching its end, we would like to kindly confirm whether we have sufficiently addressed all of your concerns (or at least some of them). Should there be any remaining questions requiring further clarification, please do not hesitate to let us know. If you are satisfied with our responses, we would greatly appreciate your consideration in adjusting the evaluation scores accordingly.
> >
> > We sincerely look forward to your feedback.

---

> > ### Comment · Reviewer_wYbH · 2025-08-07
> >
> > Thanks for your detailed rebuttal. I'll update my ratings accordingly.

---

### Author Response · Authors · 2025-08-04
**Looking Forward to the Discussion**

Dear Reviewers,

We sincerely appreciate the time and effort you’ve dedicated to reviewing our work, and we are grateful for your valuable feedback. We would like to ensure that our rebuttal has adequately addressed your concerns. As there is still time before the end of the discussion phase, we would greatly appreciate the opportunity to engage in further dialogue with you to see if there are any additional questions or points you’d like to discuss.

We look forward to hearing from you, and thank you again for your thoughtful consideration.

Best regards,

Authors

---

### Author Response · Authors · 2025-08-09
**General Response**

We thank again for all reviewers and the ACs for their hard work in NeurIPS 2025. We are encouraged by the following positive feedback:

- Discovered a significant, timely reliability problem in code reasoning beyond standard code-gen benchmarks (**wYbH, 5hDL**).
- Original, valuable benchmark with systematic, well-motivated perturbations (**wYbH, 5hDL**).
- Notable empirical insights include "reasoning collapse" and distinct MDC vs. MPS failure modes (**wYbH, dLhF**).
- Thorough analysis with broad and fine-grained comparisons (**5hDL, RETr**).
- Writing, figures, and tables are clear and easy to follow (**wYbH, 5hDL**).

During rebuttal, we made the following updates and additions to strengthen the paper:

1. Added JavaScript experiments showing the generalizability (**wYbH, 5hDL**).
2. Stress-tested with industrial JS obfuscation. Our perturbations reach hard obfuscation impact (**wYbH**).
3. Reframed NL reliance issues and reasoning collapse as failures of critical evaluation and cognitive dissonance (**wYbH**).
4. Evaluated mitigation strategies: a refactoring agent (helps with structural perturbations but not with misleading perturbations) and MCP agents (**wYbH, dLhF**).
5. Added a shuffle ablation experiment, separating reasoning from memorization (**RETr**).
6. Summarized our CoT "execution-trace" recipe and contrasted behaviors under MDC vs. MPS (**RETr**).
7. Clarified positioning versus prior robustness work and added structure-aware references (**dLhF, RETr**).
8. Improved presentation: moved back the core tables and illustration figure to the main text, and highlighted the novel findings in Section 4.3.1 and 4.4 (**RETr, dLhF**).

These suggestions provide stronger cross-language evidence, deeper analysis of failure modes, clearer novelty and positioning, and improved readability. We appreciate the reviewers' guidance. Your feedback enhanced the rigor, clarity, and impact of CodeCrash.

---

### Decision · Program_Chairs · 2025-09-17

**Decision:**

Accept (poster)

**Comment:**

Thanks for submitting to NeurIPS 2025. The manuscript introduces CodeCrash, a benchmark for code LLM stress testing under structural and textual distractions. The benchmark introduces 7 main types of distractions for code, spanning across code structural and textural domains, and expands beyond the scope of existing code robustness studies. By evaluating on mainstream commercial and open source models, the study reveals a few interesting and timely findings such as the over-reliance of models on textual/comment information, and late-minute distraction by text in CoT/reasoning models. After the discussion phase, all reviewers are satisfied with the response and additional studies that resolve major concerns. Reviewers agree on borderline acceptance with consensus due to the timely findings for the community. However, there are some remaining concerns about lack of novelty, in-depth mechanistic explanation, and the limited number of studied subject models. Overall, we are happy to see this work at the conference. Congratulations on the acceptance. Authors should ensure that they incorporate the feedback and responses in their revisions, particularly in-depth analysis.